# An ultra-short-acting benzodiazepine in thalamic nucleus reuniens undermines fear extinction via intermediation of hippocamposeptal circuits
Hoiyin Cheung[1,2,3,7], Tong-Zhou Yu[4,7], Xin Yi[2,4,7], Yan-Jiao Wu[2], Qi Wang[2], Xue Gu[2], Miao Xu[4], Meihua Cai[1], Wen Wen[1], Xin-Ni Li[4], Ying-Xiao Liu[4], Ying Sun[1], Jijian Zheng[1], Tian-Le Xu [1,2,5], Yan Luo [3] ✉, Ma-Zhong Zhang [1] ✉ & Wei-Guang Li [2,4,6] ✉

Benzodiazepines, commonly used for anxiolytics, hinder conditioned fear extinction, and the underlying circuit mechanisms are unclear. Utilizing remimazolam, an ultra-short-acting benzodiazepine, here we reveal its impact on the thalamic nucleus reuniens (RE) and interconnected hippocamposeptal circuits during fear extinction. Systemic or RE-specific administration of remimazolam impedes fear extinction by reducing RE activation through A type GABA receptors. Remimazolam enhances long-range GABAergic inhibition from lateral septum (LS) to RE, underlying the compromised fear extinction. RE projects to ventral hippocampus (vHPC), which in turn sends projections characterized by feed-forward inhibition to the GABAergic neurons of the LS. This is coupled with long-range GABAergic projections from the LS to RE, collectively constituting an overall positive feedback circuit construct that promotes fear extinction. RE-specific remimazolam negates the facilitation of fear extinction by disrupting this circuit. Thus, remimazolam in RE disrupts fear extinction caused by hippocamposeptal intermediation, offering mechanistic insights for the dilemma of combining anxiolytics with extinction-based exposure therapy.

Fear extinction, as a form of inhibitory learning and memory, enables organisms to adapt to changing environments[1–3]. It is considered a fundamental process that underlies behavioral changes in treating mental disorders, including anxiety and post-traumatic stress disorder (PTSD)[4,5]. Exposure therapy, based on the principles of fear extinction, is widely used in managing these psychiatric conditions. Simultaneously, benzodiazepines are one of the most commonly prescribed anxiolytic drugs for patients with PTSD and other conditions[6–8]. However, their use in combination with exposure therapy has produced varied clinical outcomes[9–13]. Although

benzodiazepines can alleviate anxiety during treatment, several studies suggest potential negative impacts on fear extinction and safety signal recognition for exposure therapy[14]. When rodents are exposed to benzodiazepines during extinction learning, it results in a shift in interoceptive (i.e., drug) context between extinction and retrieval test that causes fear renewal[15]. Additionally, benzodiazepines have been shown to lower the level of fear response during extinction learning when the conditioned stimulus is presented alone, which undermines the prediction error necessary for the development of new learning to counteract the conditioned fear[16–18]. The

[1]Center for Brain Science, Department of Anesthesiology and Pediatric Clinical Pharmacology Laboratory, Shanghai Children's Medical Center, National Children's Medical Center, Shanghai Jiao Tong University School of Medicine, Shanghai 200127, China. [2]Department of Anatomy and Physiology, Shanghai Jiao Tong University School of Medicine, Shanghai 200025, China. [3]Department of Anesthesiology, Ruijin Hospital, Shanghai Jiao Tong University School of Medicine, Shanghai 200025, China. [4]Department of Rehabilitation Medicine, Huashan Hospital, Institute for Translational Brain Research, State Key Laboratory of Medical Neurobiology and Ministry of Education Frontiers Center for Brain Science, Fudan University, Shanghai 200032, China. [5]Songjiang Hospital and Songjiang Research Institute, Shanghai Key Laboratory of Emotions and Affective Disorders, Shanghai Jiao Tong University School of Medicine, Shanghai 201600, China. [6]Ministry of Education-Shanghai Key Laboratory for Children's Environmental Health, Xinhua Hospital, Shanghai Jiao Tong University School of Medicine, Shanghai 200092, China. [7]These authors contributed equally: Hoiyin Cheung, Tong-Zhou Yu, Xin Yi. ✉e-mail: ly11087@rjh.com.cn; zmzscmc@shsmu.edu.cn; liwg@fudan.edu.cn

anxiolytic effects of benzodiazepines can be intertwined with their sedative, hypnotic, and anesthetic effects[6,19], necessitating the use of ultra-short-acting benzodiazepines to delineate their impact on fear extinction while minimizing interference with interoceptive states.

Remimazolam, a novel benzodiazepine derivative, has been developed to enhance sedation profiles. It presents a quicker onset, a shorter sedation duration, and faster recovery when compared to current agents[20–23]. The ester-metabolized structure of remimazolam contributes to its remarkable effectiveness in procedural sedation, highlighting its versatility and effectiveness in clinical settings[24–27]. By utilizing remimazolam as an ultra-short-acting benzodiazepine, we aimed to investigate the effects and underlying mechanisms of benzodiazepines on fear extinction, with an emphasis on circuit mechanisms to gain mechanistic insights into how to reconcile the use of anxiolytic drugs with extinction-based exposure therapy for patients with PTSD.

From a circuit perspective, fear extinction relies on the interactive engagement of a tripartite circuit involving the ventral hippocampus (vHPC), medial prefrontal cortex (mPFC), and basolateral amygdala (BLA)[1–3,28–30]. The nucleus reuniens (RE) of the midline thalamus not only sends dense projections to the hippocampal CA1 region, but also receives dense projections from the mPFC[31–33]. It is thus important for interconnecting the mPFC and hippocampus for fear extinction[34–36]. RE dynamically engages in oscillatory activities, coupling with the mPFC and hippocampus during sleep and wakefulness to regulate emotional learning and memory[37–42]. Notably, RE coordinates prefrontal-hippocampal synchrony to suppress fear memories post-extinction[43]. Yet, whether RE, with its state-dependent neuronal dynamics, could be a potential target for benzodiazepine modulation of fear extinction remains unexplored[34,42,43]. Furthermore, while the excitatory inputs from the mPFC to RE in fear extinction are understood, further research is required to unravel the inhibitory inputs to RE and their potential role in fear extinction.

Leveraging remimazolam, an ultra-short-acting benzodiazepine, this study examined its impact on fear extinction. By investigating the effects of remimazolam on fear extinction-related circuits, with an emphasis on the RE, the present study revealed that remimazolam as a positive modulator of A type GABA receptors (GABA$_A$Rs) potentiated GABAergic transmission in the RE. Additionally, it decreased neuronal excitability of RE and altered the hippocamposeptal circuit function. This modification elucidated the benzodiazepine modulation of fear extinction, offering mechanistic insights to address the challenge of integrating anxiolytics with extinction-based exposure therapy.

## Results
### Systemic remimazolam treatment hinders fear extinction
The investigation into the pharmacological effects of remimazolam on fear extinction began with a meticulous screening of dosages to ensure its impact on extinction efficacy without influencing overall locomotor activity post-injection. Leveraging previous reports highlighting the ultra-short-acting nature of remimazolam[20], we established the 15-min time frame as critical for assessing its action. Intraperitoneal (i.p.) administration of remimazolam with doses ranging from 2 to 8 mg/kg was conducted, revealing that doses up to 4 mg/kg did not affect overall locomotor activity in the open field test, while higher doses did (Supplementary Fig. 1a–f). To ensure unaffected detection of the conditioned freezing behavior, 2 and 4 mg/kg were selected for subsequent fear extinction studies.

The auditory fear conditioning paradigm was employed to assess the effects of remimazolam on freezing behavior during fear extinction learning and retrieval (Fig. 1a and Supplementary Fig. 1g). On day 1 of fear conditioning, mice underwent individual placement in a conditioning chamber (context A) and were exposed to five pure tones (CS), each paired with foot shocks. This elicited a gradual increase in freezing during conditioning trials, indicating successful acquisition of conditioned fear (Fig. 1b, left). Subsequently, mice were tested in a novel context (context B) and exposed to the cued tone (CS) alone, without foot shock, for 12 trials. Systemic administration of remimazolam at a dose of 4 mg/kg i.p., but not at 2 mg/kg,

significantly impeded fear extinction (Fig. 1b, c and Supplementary Fig. 1h). This was evidenced by a slower decrease in conditioned freezing responses during extinction learning on day 2 compared to vehicle control mice (Fig. 1b, c). Moreover, on day 3, remimazolam-treated mice exhibited persistently higher freezing levels during extinction retrieval (Retr.) (Fig. 1b, c). Notably, in addition to undermining fear extinction, remimazolam also seemed to facilitate the expression of conditioned fear, as remimazolam-injected mice exhibited higher levels of freezing during the initial CS presentations of extinction (Fig. 1b and Supplementary Fig. 1h). However, remimazolam did not impact freezing levels in mice subjected to similar conditioning and extinction protocols but without foot shocks (CS only, Supplementary Fig. 1j–l). Overall, these results suggest that administering remimazolam before initial extinction learning hinders extinction learning and disrupts subsequent extinction retrieval sessions.

### Remimazolam decreases neuronal activity in the RE and vHPC while increasing it in the LS during extinction learning
To pinpoint the specific cell populations responsible for the impaired fear extinction induced by remimazolam, we conducted immunohistochemistry to assess the expression of the activity-induced immediate-early gene, c-fos, 90 min after day 3 of extinction retrieval. A comparison of two groups—those treated with vehicle control or remimazolam prior to extinction learning—across the whole brain revealed significant decreases in neuronal activation in the RE and vHPC (Fig. 1d–f). Notably, quantification data pertaining to c-fos-positive neurons along the anterior-posterior axis of the RE demonstrated that the anterior RE exhibits a more pronounced response to remimazolam compared to the posterior RE (Fig. 1f). Furthermore, remimazolam pretreatment also elicited increased activation in the lateral septum (LS), including its dorsal part (LSD), ventral part (LSV), and intermediate part (LSI) (Fig. 1d–f). These results suggest that remimazolam induces a reconfiguration of neuronal activation in distinct brain regions associated with fear extinction.

### RE-specific remimazolam delivery impedes fear extinction
The study then assessed whether remimazolam specifically targeted to the RE, a key region implicating in fear extinction[34,42,43], would be sufficient to hinder fear extinction. In this experiment, remimazolam (100 μM, 0.5 μl, with a single central injection allowing bilateral infusion) or vehicle control (artificial cerebrospinal fluid, ACSF) was infused into the RE (Fig. 1g, h) 15 min prior to fear extinction learning on day 2. Subsequently, on day 3, mice then underwent extinction retrieval. Remarkably, mice treated with remimazolam exhibited significantly higher freezing levels than those treated with the vehicle during both extinction learning and retrieval (Fig. 1i, j). As a control for locomotor activity, RE-specific remimazolam delivery did not affect behaviors in the open field test (Supplementary Fig. 2). These results underscore the role of remimazolam on RE activity in undermining fear extinction, emphasizing the specificity of its potential effects in the RE.

### Remimazolam enhances electrically-evoked GABAergic inhibitory postsynaptic currents and reduces neuronal excitability in RE neurons
To elucidate the pharmacological impact of remimazolam on the RE, we investigated its effects on GABAergic inhibition by measuring electrically-evoked inhibitory postsynaptic currents (eIPSCs) in brain slices containing the RE (Fig. 2a). Remimazolam (100 μM) significantly increased the amplitudes of eIPSCs by 100% compared to the control condition and prolonged their decay time (Fig. 2b–d), as confirmed by the GABA$_A$R antagonist, picrotoxin (PTX, 100 μM). These results indicate that remimazolam enhances GABAergic transmission in the RE. In contrast, remimazolam did not affect the electrically evoked excitatory postsynaptic currents (eEPSCs), as validated by antagonists for glutamatergic AMPA and NMDA

**Fig. 1 | Remimazolam inhibits fear extinction and alters the activity of associated brain areas.**
**a**, **d**, **g** Experimental design. Either systemic or RE-specific remimazolam (4 mg in 5 ml saline per kg body weight of mice, i.p., **a**, **d**; 100 μM, 0.5 μl in ACSF, with a single central intracerebral injection allowing bilateral infusion, **g**) were given 15 min before fear extinction. **h** *Left*, Schematics of cannula implantation. *Right*, Representative image of implantation sites in RE. Scale bar, 200 μm.
**b**, **c**, **i**, **j** Time course of freezing responses to the CS.
**c**, **j** Freezing responses during extinction retrieval.
**b**, **c** Vehicle group, n = 7 mice; Remimazolam group, n = 11 mice. **b** Statistics: two-way repeated measures ANOVA, Cond.: $F_{(1, 16)} = 0.0002063$, $P = 0.9887$; Ext.: $F_{(1, 16)} = 27.64$, $^{***}P < 0.0001$; Retr.: $F_{(1, 16)} = 34.91$, $^{***}P < 0.0001$. **c**, Statistics: two-tailed unpaired Student's *t*-test, $t_{(16)} = 5.909$, $^{***}P < 0.0001$. **i**, **j** Vehicle group, n = 7 mice; Remimazolam group, n = 9 mice. **i** Statistics: two-way repeated measures ANOVA, Cond.: $F_{(1, 14)} = 0.01148$, $P = 0.9162$; Ext.: $F_{(1, 14)} = 20.59$, $^{***}P = 0.0005$; Retr.: $F_{(1, 14)} = 7.195$, $^{*}P = 0.0179$. **j** Statistics: two-tailed unpaired Student's *t* test, $t_{(14)} = 2.682$, $^{*}P = 0.0179$.
**e** Representative images of c-fos$^+$ (red) cell immunofluorescence. Scale bar, 200 μm. **f** Quantification of c-fos positive neurons. Statistics: two-tailed unpaired Student's *t* test, $^{*}P < 0.05$, $^{**}P < 0.01$, $^{***}P < 0.001$. n = 4 mice per group. PL prelimbic cortex; IL infralimbic cortex; Cg cingulate cortex; AID agranular insular cortex, dorsal part; AIV agranular insular cortex, ventral part; GI granular insular cortex; DI dysgranular insular cortex, S1 primary somatosensory cortex, Acbsh accumbens nucleus shell, Acbc accumbens nucleus, core, LSD lateral septal nucleus, dorsal part, LSV lateral septal nucleus, ventral part, LSI lateral septal nucleus, intermediate part, MS medial septal nucleus, PVA paraventricular thalamic nucleus, anterior part, PT paratenial thalamic nucleus, IAM interanteromedial thalamic nucleus, IMD intermediodorsal thalamic nucleus, RE reuniens thalamic nucleus, Vre ventral reuniens thalamic nucleus, Xi xiphoid thalamic nucleus, Rh rhomboid thalamic nucleus, Sub submedius thalamic nucleus, dHPC hippocampus, dorsal part, vHPC hippocampus, ventral part, CeA central amygdaloid nucleus, BLA basolateral amygdaloid nucleus, anterior part, PAG periaqueductal gray, DA dorsal hypothalamic area, Cl claustrum, DEn dorsal endopiriform nucleus. Data are presented as mean ± SEM.

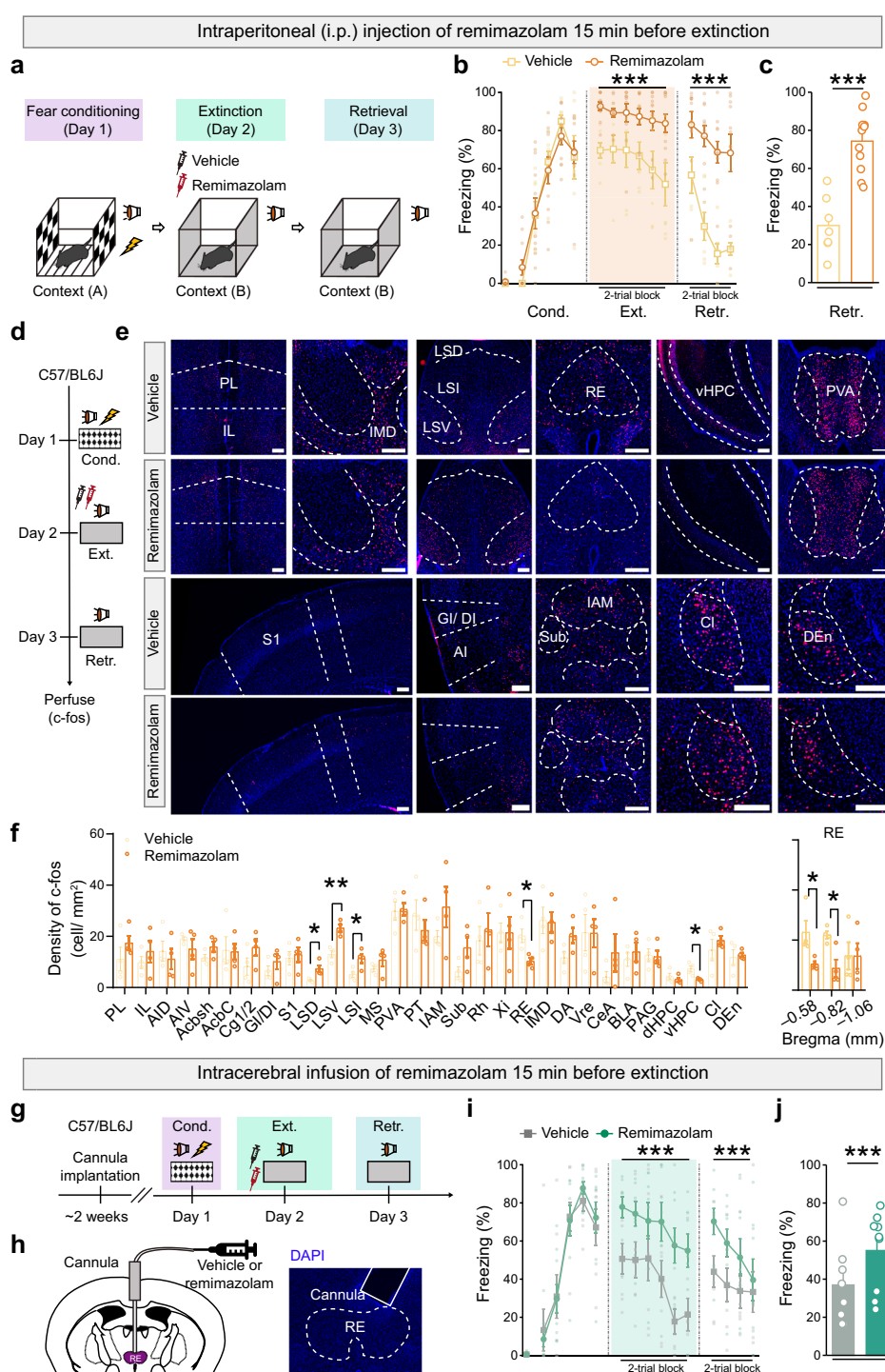

receptors, CNQX (20 μM) and APV (50 μM). These results demonstrate the specificity for remimazolam's regulation of GABA$_A$Rs (Fig. 2e–g).

To further explore the impact of remimazolam on RE neuronal excitability during fear extinction, we also recorded the evoked spiking of RE neurons in response to step-depolarization current injection. Remimazolam (100 μM) significantly increased the rheobase required for all-or-none firing and reduced depolarization-induced spike firing of neurons (Fig. 2h–k). These results align with the promoting effects of remimazolam on RE GABAergic inhibition and are consistent with its observed hindrance of fear extinction.

## Effects of remimazolam are associated with GABA$_A$R activity in RE

Subsequently, we investigated whether the impact of remimazolam on fear extinction is linked to the activity of GABA$_A$Rs, aiming to connect its behavioral effects with the observed potentiation of GABAergic transmission at the synaptic level (Fig. 2). Initially, we confirmed that the application of the benzodiazepine antagonist, flumazenil (100 μM), nullified the effects of remimazolam on the eIPSCs in RE, while leaving the eEPSCs unaffected (Supplementary Fig. 3a–e). Behaviorally, systemic pretreatment with flumazenil (4 mg/kg, i.p., 15 min before administration of remimazolam) mitigated the effects of RE-specific remimazolam delivery on fear extinction

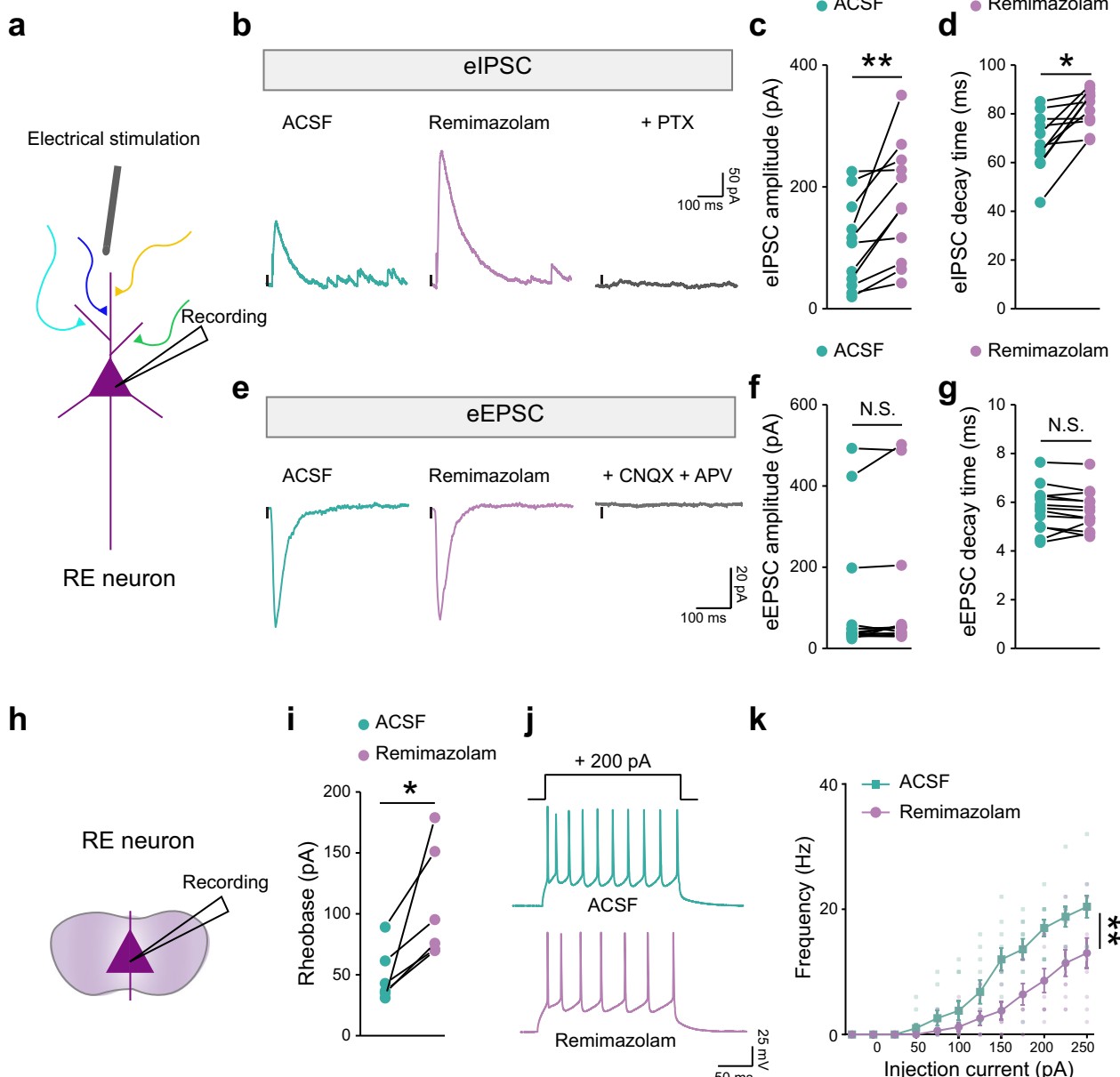

**Fig. 2 | Effects of remimazolam on synaptic transmission and excitability of RE neurons. a, h** Schematic diagram of whole-cell patch clamp recording. Example traces of eIPSCs (**b**, holding = 0 mV) and eEPSCs (**e**, holding = –70 mV). Statistical analysis of the amplitude (**c, f**) and decay time (**d, g**) of eIPSCs and eEPSCs. Statistics: two-tailed paired Student's *t* test, (**c**) $t_{(10)} = 3.657$, $^{**}P = 0.0044$; (**d**) $t_{(10)} = 3.888$, $^{*}P = 0.0179$, n = 11 cells; (**f**) $t_{(12)} = 1.612$, $P = 0.1330$; (**g**) $t_{(12)} = 0.6396$, $P = 0.5344$. n = 13 cells. **i** Statistical analysis of the rheobase of ramping current injection-induced action potentials. Statistics: two-tailed paired Student's *t* test, $t_{(5)} = 3.084$, $^{*}P = 0.0273$. n = 6 cells. **j** Example traces of stepping current injection induced action potentials treated before (*Left*) and after (*Right*) bath application of remimazolam (100 μM). **k** The frequency of action potential discharge as a function of step-current intensity (–25 to 250 pA, 500 ms). Statistics: two-way repeated measures ANOVA, $F_{(1, 18)} = 10.48$, $^{**}P = 0.0046$. n = 10 cells. Data are presented as mean ± SEM.

(Supplementary Fig. 3f–i). These results support the notion that remimazolam undermines fear extinction by targeting the benzodiazepine sites on GABA$_A$Rs, consistent with previously characterized molecular pharmacological profiles[20].

Our focus then shifted to the γ2 subunit of GABA$_A$Rs, which is present in approximately 90% of GABA$_A$Rs[6]. We injected the AAV-CAG-EGFP-U6-Gabrg2-ShRNA virus targeting the γ2 subunit of GABA$_A$Rs (*Gabrg2*), alongside the negative control (NC) virus (AAV-CAG-EGFP-U6-NC)[44], into the RE of different groups of mice (Fig. 3a, b). AAV-CAG-EGFP-U6-Gabrg2-ShRNA notably reduced the protein level of γ2 subunit of GABA$_A$Rs in the RE (Supplementary Fig. 4a–e) and unexpectedly caused a tendency to diminish the excitability of RE neurons (rheobase, $t_{(13)} = 1.372$, $P = 0.1933$, n = 7 to 8 cells per group, NC vs. Gabrg2-ShRNA, unpaired

Student's *t*-test; depolarization-induced spike firing, $F_{(1, 14)} = 3.728$, $P = 0.0740$, n = 7 to 9 cells per group, NC *vs.* Gabrg2-ShRNA, two-way repeated measures ANOVA; data from Fig. 3c–l). This paradoxical hypoexcitability, despite unreliable inhibition due to nonfunctional GABA$_A$Rs, likely involves homeostatic synaptic scaling, which adjusts synaptic strength in response to prolonged changes of neuronal activity[45,46]. Behaviorally, mice injected with AAV-CAG-EGFP-U6-gabag2-ShRNA displayed significantly higher levels of freezing during fear conditioning, extinction learning and retrieval compared to those injected with AAV-CAG-EGFP-U6-NC (Supplementary Fig. 4f–i). These findings support the negative correlation between RE neuronal excitability and the behavioral expression of conditioned fear, aligning well with the physiological role of RE in fear extinction.

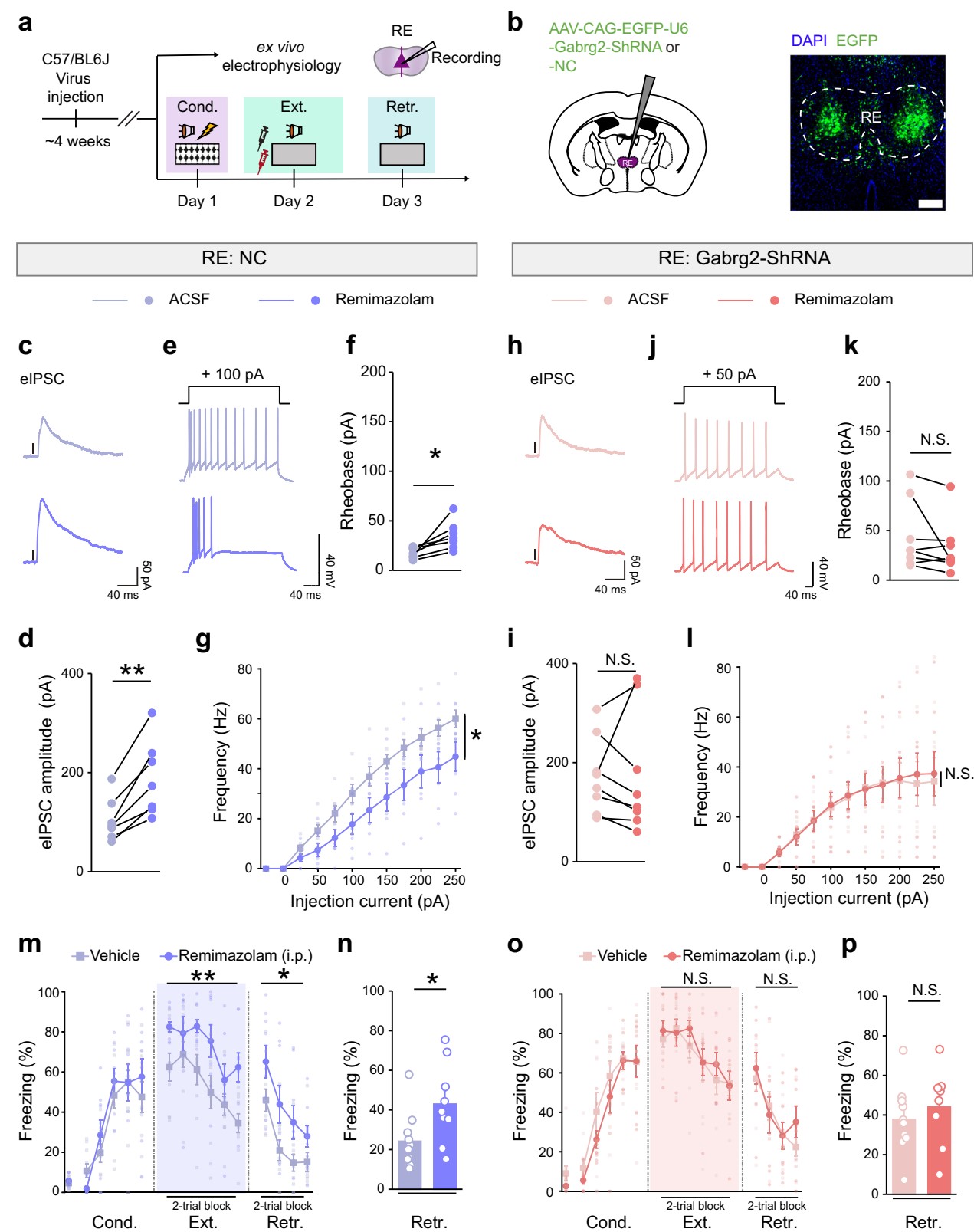

Further examination of freezing levels between vehicle and remimazolam for each viral condition corroborated the necessity of GABA$_A$Rs in RE for the responsiveness to remimazolam, both electrophysiologically and behaviorally. In brain slices from AAV-CAG-EGFP-U6-NC-injected mice, remimazolam significantly enhanced eIPSCs and reduced evoked spiking of RE neurons in response to step-depolarization current injection (Fig. 3c–g).

Conversely, in RE neurons of brain slices from AAV-CAG-EGFP-U6-Gabrg2-ShRNA-injected mice, remimazolam had no effects on eIPSCs or neuronal spiking (Fig. 3h–l). Behaviorally, in AAV-CAG-EGFP-U6-Gabrg2-ShRNA-injected mice, systemic administration (i.p.) of remimazolam did not influence cued freezing during extinction learning and retrieval compared to the AAV-CAG-EGFP-U6-NC group (Fig. 3m–p).

**Fig. 3 | Effects of remimazolam on fear extinction in mice with RE-specific genetic knockdown of γ2-GABA$_A$R (Gabrg2) subunit. a** Experimental design. **b** *Left*, Schematics of AAV injections. *Right*, Representative image of EGFP expression in RE. Scale bar, 200 μm. **c–g, h–l** Whole-cell patch clamp recordings. RE-NC group, n = 7 cells; RE-Gabrg2-ShRNA group, n = 8 cells. **c, h** Example traces of eIPSCs (holding = 0 mV). **d, i** Statistics: two-tailed paired Student's *t* test, **d** $t_{(6)}$ = 5.364, $^{**}P$ = 0.0017; **i** $t_{(7)}$ = 0.05885, P = 0.9547. **e, j** Example traces of stepping current injection-induced action potentials before (*Upper*) and after (*Lower*) application of remimazolam (100 μM). **f, k** Statistical analysis of the rheobase of ramping current injection-induced action potentials. Statistics: two-tailed paired Student's *t* test, **f**, $t_{(6)}$ = 3.177, $^{*}P$ = 0.0192; (**k**) $t_{(7)}$ = 1.790, P = 0.1166. The frequency of AP discharge as a function of step-current intensity (−25 to 250 pA, 500 ms). Statistics:

two-way repeated measures ANOVA, **g** $F_{(1, 12)}$ = 5.263, $^{*}P$ = 0.0406; **l** $F_{(1, 16)}$ = 0.01144, P = 0.9162. **m–p** Effects of remimazolam (4 mg in 5 ml saline per kg body weight of mice, i.p.) on fear extinction in mice with RE expressing NC (**m, n**, Vehicle group, n = 10 mice; remimazolam group, n = 9 mice) or Gabrg2-ShRNA (**o** and **p**, Vehicle group, n = 10 mice; remimazolam group, n = 9 mice). Time course of freezing responses to the CS. Statistics are as follows: two-way repeated measures ANOVA, **m** Cond.: $F_{(1, 17)}$ = 0.2737, P = 0.6076; Ext.: $F_{(1, 17)}$ = 8.478, $^{**}P$ = 0.0097; Retr.: $F_{(1, 17)}$ = 5.692, $^{*}P$ = 0.0289. **o** Cond.: $F_{(1, 17)}$ = 1.117, P = 0.3054; Ext.: $F_{(1, 17)}$ = 0.1975, P = 0.6624; Retr.: $F_{(1, 17)}$ = 0.1346, P = 0.7182. Freezing responses during extinction retrieval. Statistics are as follows: two-tailed unpaired Student's *t*-test, **n** $t_{(17)}$ = 2.386, $^{*}P$ = 0.0289; **p** $t_{(17)}$ = 0.3669, P = 0.7182. Data are presented as mean ± SEM.

These results underscore the specific involvement of RE GABA$_A$R activity in the modulation of fear extinction by remimazolam.

## GABAergic inhibition on RE-vHPC projectors largely confers remimazolam modulation on fear extinction

To dissect the circuit mechanisms within the RE contributing to the modulation of fear extinction by remimazolam, we conducted a cell-type-specific rescue experiment. This involved co-introducing a Cre-dependent exogenous *Gabrg2* into distinct subpopulations of RE projection neurons alongside Gabrg2-shRNA (AAV-U6-Gabrg2-ShRNA-DIO-Gabrg2*-EGFP) (Fig. 4a). Considering the dense excitatory projections of RE toward the hippocampal CA1 region[31–33], particularly its ventral segment (see below for electrophysiological characterization) crucial for fear extinction[28–30], we probed whether the RE-to-vHPC projection underlies the remimazolam's effects on fear extinction. The AAV vector carrying Gabrg2-shRNA also featured a double-floxed inverted orientation (DIO) coding sequence for shRNA-resistant *Gabrg2-EGFP* (read as *Gabrg2*\*), expressed exclusively in the presence of Cre recombinase (Fig. 4b). Co-injection into the vHPC with retrograde AAV expressing Cre (Retro-AAV-Syn-Cre-mCherry), where Cre recombinase is expressed in RE-vHPC projectors, led to robust expression of Gabrg2-EGFP (Fig. 4b) in a subset of RE neurons that targets to the vHPC region. As a control, co-injection with retrograde AAV expressing mCherry only (Retro-AAV-Syn-mCherry), lacking Cre recombinase expression in RE-vHPC projectors, showed no expression of Gabrg2-EGFP in RE neurons. The shRNA within this AAV effectively abolished remimazolam responsiveness in mice injected with the control Retro-AAV-Syn-mCherry. Conversely, in mice injected with Retro-AAV-Syn-Cre-mCherry, the virus reinstated remimazolam responsiveness. Electrophysiologically, remimazolam had no effect on the excitability of RE neurons in the control Retro-AAV-Syn-mCherry group (Fig. 4c–e) but significantly reduced the excitability of RE neurons in the Retro-AAV-Syn-Cre-mCherry group (Fig. 4f–h). Behaviorally, systemic administration (i.p.) of remimazolam had no impact on fear extinction in the control Retro-AAV-Syn-mCherry group (Fig. 4i, j) but hindered extinction learning and disrupted subsequent retrieval sessions in the Retro-AAV-Syn-Cre-mCherry group (Fig. 4k, l). Consistently, chemogenetic activation of vHPC neurons innervated by the RE also abolished the remimazolam responsiveness on fear extinction behaviors (Supplementary Fig. 5). These results collectively underscore the pivotal role of RE-vHPC projectors in mediating the regulatory effects of remimazolam on fear extinction.

## Remimazolam potentiates long-range GABAergic inhibition from LS to RE-vHPC projectors, hindering fear extinction

To delineate the monosynaptic inputs to RE, we employed retrograde tracing by injecting retrograde AAV expressing EGFP (Retro-AAV-Syn-EGFP) into the RE of wild-type mice (Fig. 5a). This viral approach enabled the retrograde labeling of neurons throughout the brain regions expressing EGFP, representing potential presynaptic inputs to RE. Neurons projecting to RE (expressing EGFP) were identified in various cortical and subcortical regions (Fig. 5b, c). Within the cortical areas, RE neurons received extensive inputs from the mPFC, encompassing both the prelimbic (PL) and infralimbic (IL) areas, as well as the cingulate cortex (Cg).

The insular cortex, particularly the agranular insular cortex (AI), dorsal (AID), and ventral (AIV) parts, but not the granular insular cortex (GI) nor dysgranular insular cortex (DI), featured prominently in these inputs. In comparison, there were relatively fewer inputs from the somatosensory cortex (S1) but more extensive inputs from the motor cortex (M1 and M2), ectorhinal cortex (ECT), auditory cortex (AU), perirhinal cortex (PRH), and lateral orbital cortex (LO) (Fig. 5b, c). These projection patterns align with previous anatomical characterizations[31] and the potential connectivity with fear extinction-related regions to account for the known role of RE in fear extinction[34]. Notably, within the thalamic areas, RE neurons received extensive inputs from the lateral septum (LS), with apparently fewer inputs from the medial septum (MS) (Fig. 5b, c). Additionally, RE neurons received substantial inputs from the midbrain periaqueductal gray (PAG), suggesting diverse roles for RE (Fig. 5b, c). Interestingly, RE neurons also received modest inputs from the dorsal and ventral hippocampi (Fig. 5b, c), implying potential signal integration between RE and the hippocampus.

Among these varied long-range inputs, we examined the specific connectivity of inputs from LS to RE, as that LS predominantly houses long-range GABAergic neurons[47] and a small population of glutamatergic neurons[48]. Moreover, in above results, we showed that systemic treatment of remimazolam enhanced LS activation during extinction retrieval (Fig. 1d–f). We speculated that the long-range GABAergic projections from LS to RE could be positively modulated by remimazolam in the scenario of fear extinction. Additionally, LS is implicated as a hub for mood, motivation, and movement[49], suggesting its potential contribution to the behavioral control of fear extinction. Whole-cell electrophysiological recordings in RE slices were conducted to assess the connectivity of inputs from LS to RE. We injected AAV-Syn-ChR2-EGFP into LS to express ChR2-EGFP in LS projection neurons together with AAV-CAG-EGFP-U6-Gabrg2-ShRNA or AAV-CAG-EGFP-U6-NC into RE to modulate the expression of GABA$_A$Rs in RE neurons (Fig. 5d). Subsequently, RE neurons were subjected to whole-cell electrophysiological recordings while LS axon terminals in RE were optogenetically activated (Fig. 5e). Remarkably, optically-evoked inhibitory post-synaptic currents (oIPSCs) were observed in RE neurons, as validated by the application of PTX (100 μM) as the GABA$_A$R antagonist (Fig. 5f, g). Notably, the oIPSCs recorded in RE neurons with AAV-CAG-EGFP-U6-NC expression were enhanced by the application of remimazolam (Fig. 5h–j). Nealy all recorded RE neurons (94.7%, 18/19) exhibited prominent oIPSCs, which were further potentiated by remimazolam (Fig. 5h–j). The quantification of oIPSC onset latencies demonstrated monosynaptic inputs from LS to RE, irrespective of the absence or presence of remimazolam (Fig. 5j). However, the oIPSCs recorded in RE neurons with AAV-CAG-EGFP-U6-Gabrg2-ShRNA expression were less pronounced (33.3%, 4/12) and unresponsive to remimazolam (Fig. 5k–m). These results indicate that LS provides substantial long-range GABAergic inhibition to RE neurons, which can be potentiated by remimazolam and likely underlie the behavioral regulation of fear extinction.

To further validate the role of LS → RE projections in fear extinction, we injected AAV-Syn-ChR2-EGFP into LS to express ChR2-EGFP in LS

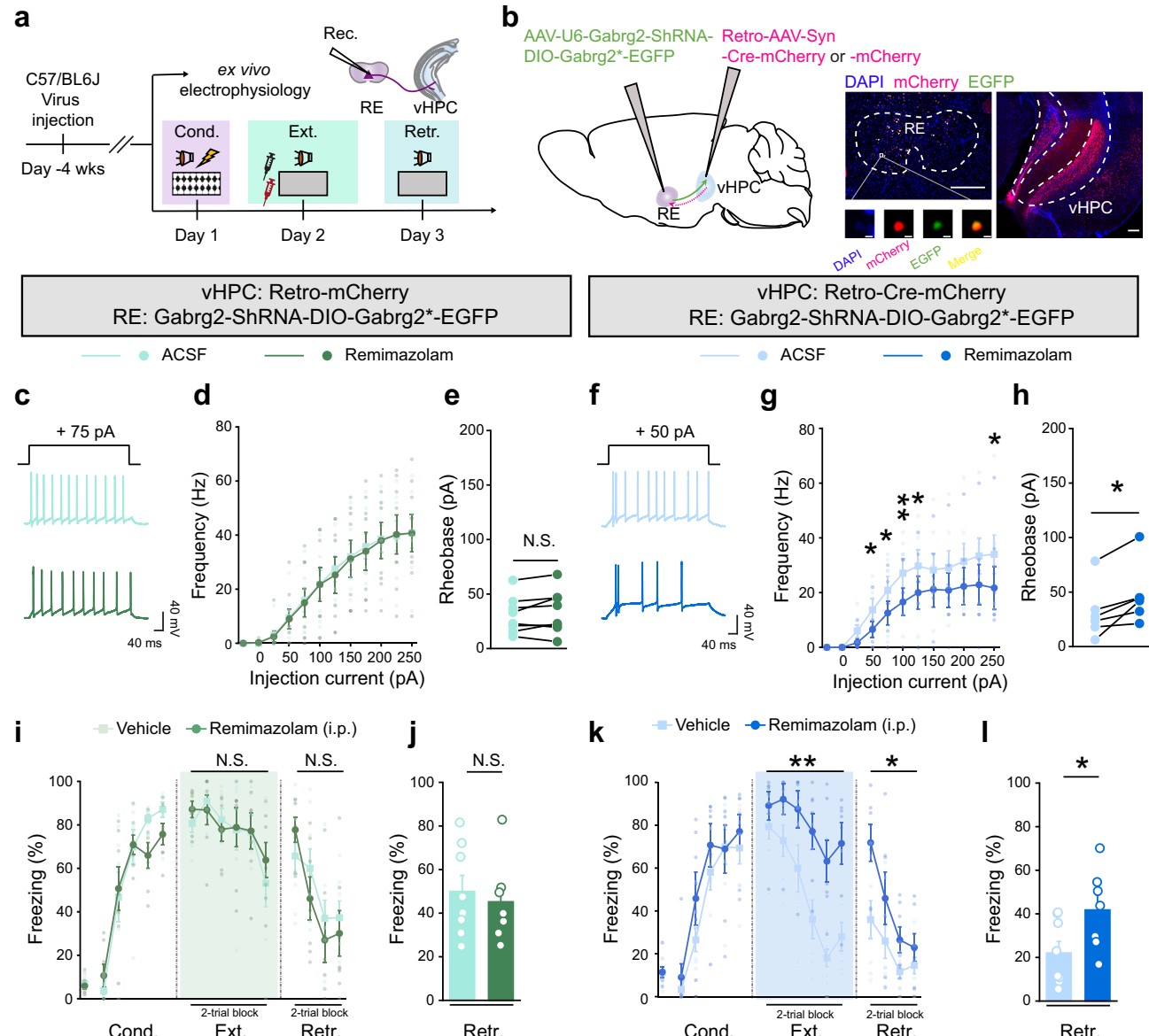

**Fig. 4 | Reinstatement of Gabrg2 expression in RE-vHPC projectors restores remimazolam responsiveness in RE Gabrg2 knockdown mice. a** Experimental design. **b** *Left*, Schematics of AAV injections. *Right*, Representative images showing the expression of EGFP (green) and mCherry (red) in RE or vHPC. Scale bar, 200 μm. The inset shows local magnification of a cell in RE (scale bar, 5 μm). **c–h** Whole-cell patch clamp recording of RE neurons. **c–e** RE Gabrg2 knockdown group, n = 8 cells. **f–h** RE Gabrg2 knockdown compensating RE-vHPC projectors with Gabrg2, n = 7 cells. **c, f** Example traces of stepping current injection induced action potentials before (*Upper*) and after (*Lower*) bath application of remimazolam (100 μM). **d, g** The frequency of AP discharge as a function of step-current intensity (−25 to 250 pA, 500 ms). Statistics are as follows: two-way repeated measures ANOVA and two-tailed unpaired Student's *t* test, (**d**) $F_{(1, 14)} = 0.009198$, $P = 0.9250$, (**i**) $F_{(1, 12)} = 1.867$, $P = 0.1968$. **e, h** Statistical analysis of the rheobase of ramping

current injection induced action potentials. Statistics: two-tailed paired Student's *t* test, **e** $t_{(7)} = 1.440$, $P = 0.1931$; **h** $t_{(5)} = 3.450$, $^*P = 0.0182$. **i–l** Effects of remimazolam (4 mg in 5 ml saline per kg body weight of mice, i.p.) on fear extinction in mice with RE Gabrg2 knockdown (**i, j** Vehicle group, n = 8 mice, remimazolam group, n = 7 mice) or those compensating RE-vHPC projectors with Gabrg2 mice (**k, l** Vehicle group, n = 8 mice, remimazolam group, n = 7 mice) on fear extinction. **i, k** Time course of freezing responses to the CS. Statistics are as follows: two-way repeated measures ANOVA, **i** Cond.: $F_{(1, 13)} = 0.3489$, $P = 0.5649$; Ext.: $F_{(1, 13)} = 0.04041$, $P = 0.8438$; Retr.: $F_{(1, 13)} = 0.2116$, $P = 0.6531$. **k** Cond.: $F_{(1, 13)} = 1.176$, $P = 0.2978$; Ext.: $F_{(1, 13)} = 10.48$, $^{**}P = 0.0065$; Retr.: $F_{(1, 13)} = 5.271$, $^*P = 0.0390$. **j, l** Freezing responses during extinction retrieval. Statistics are as follows: two-tailed unpaired Student's *t* test, **j** $t_{(13)} = 0.4600$, $P = 0.6531$; **l** $t_{(13)} = 2.296$, $^*P = 0.0390$. Data are presented as mean ± SEM.

projection neurons and implanted optic fibers into RE, enabling optogenetic activation of LS → RE projections (Fig. 5n, o). Remarkably, in comparison to the control group injected with AAV-Syn-EGFP, optogenetic activation of LS → RE pathways significantly hindered fear extinction (Fig. 5p–r), mirroring the effects of remimazolam. As a control for locomotor activity, optogenetic activation of LS → RE pathways did not affect the behaviors in open field test (Supplementary Fig. 6). Thus, remimazolam has the capacity to enhance the GABAergic LS → RE projection, thereby reducing fear extinction.

## Activation of the RE → vHPC → LS pathway promotes fear extinction, with reversal by inhibiting RE-vHPC projectors

After establishing the pivotal role of RE-vHPC projectors in mediating the effects of remimazolam (Fig. 4) and identifying LS as a source of long-range GABAergic inhibition to RE, we delved into the synaptic inputs to LS involved in regulating fear extinction. Leveraging the well-documented existence of hippocamposeptal circuits[49], we investigated whether vHPC neurons, receiving excitatory inputs from RE, project their axons to LS for fear extinction control. To explore this, we injected anterograde

**Fig. 5 | Remimazolam enhances long-range GABAergic inhibition from LS to RE-vHPC projectors, impairing fear extinction. a, d** Schematic of AAV injections. **b** Representative images. Scale bar, 200 µm. **c** Quantification of EGFP⁺ neurons. n = 4 mice. PL prelimbic cortex; IL infralimbic cortex; Cg cingulate cortex; AID agranular insular cortex, dorsal part; AIV agranular insular cortex, ventral part; DI dysgranular insular cortex; GI granular insular cortex; S1 primary somatosensory cortex; M1 primary motor cortex; M2 secondary motor cortex; ECT ectorhinal cortex; AU auditory cortex; PRH perirhinal cortex; LO lateral orbital cortex; LSD lateral septal nucleus, dorsal part; LSI lateral septal nucleus, intermediate part; LSV lateral septal nucleus, ventral part; PAG periaqueductal gray. **e** Schematic of whole-cell patch clamp recording. **f** Example traces of oIPSC at LS → RE projections before (*Upper*) and after (*Lower*) bath application of picrotoxin (PTX, 100 µM). **g** Statistical analysis. Statistics are as follows: two-tailed paired Student's *t* test, $t_{(2)}$ = 5.742, *P = 0.0290, n = 3 cells. **h–m** Example traces of oIPSC at LS → RE projections (**i, l**) and statistical analysis of the percentage of cells with oIPSC (**h, k**) and the amplitude of oIPSC (**j, m**) before and after bath application of remimazolam (100 µM). Statistics are as follows: two-tailed paired Student's *t* test, **j**, *Left*, $t_{(6)}$ = 2.806, *P = 0.0309; *Right*, $t_{(6)}$ = 0.000, *P > 0.9999, n = 7 cells; **m** $t_{(6)}$ = 0.9343, P = 0.3862, n = 7 cells. **n** Experimental design. **o** Schematics of AAV injections (*Upper*) and representative images (*Lower*) of EGFP expression (green) and fiber implantations in LS and RE, respectively. Scale bar, 200 µm. **p** Example traces of action potentials of RE neurons evoked by 473 nm blue light at a frequency of 5 Hz, 10 Hz, and 20 Hz. **q, r** Effects of activation of LS terminals in RE on fear extinction. EGFP group, n = 7 mice; ChR2 group, n = 8 mice. **q** Time course of freezing responses to the CS. Statistics are as follows: two-way repeated measures ANOVA, Cond.: $F_{(1, 13)}$ = 2.930, P = 0.1107; Ext.: $F_{(1, 13)}$ = 0.7261, P = 0.4096; Retr.: $F_{(1, 13)}$ = 6.239, *P = 0.0267. **r** Freezing responses during extinction retrieval. Statistics are as follows: two-tailed unpaired Student's *t*-test, $t_{(13)}$ = 2.498, *P = 0.0267. Data are presented as mean ± SEM.

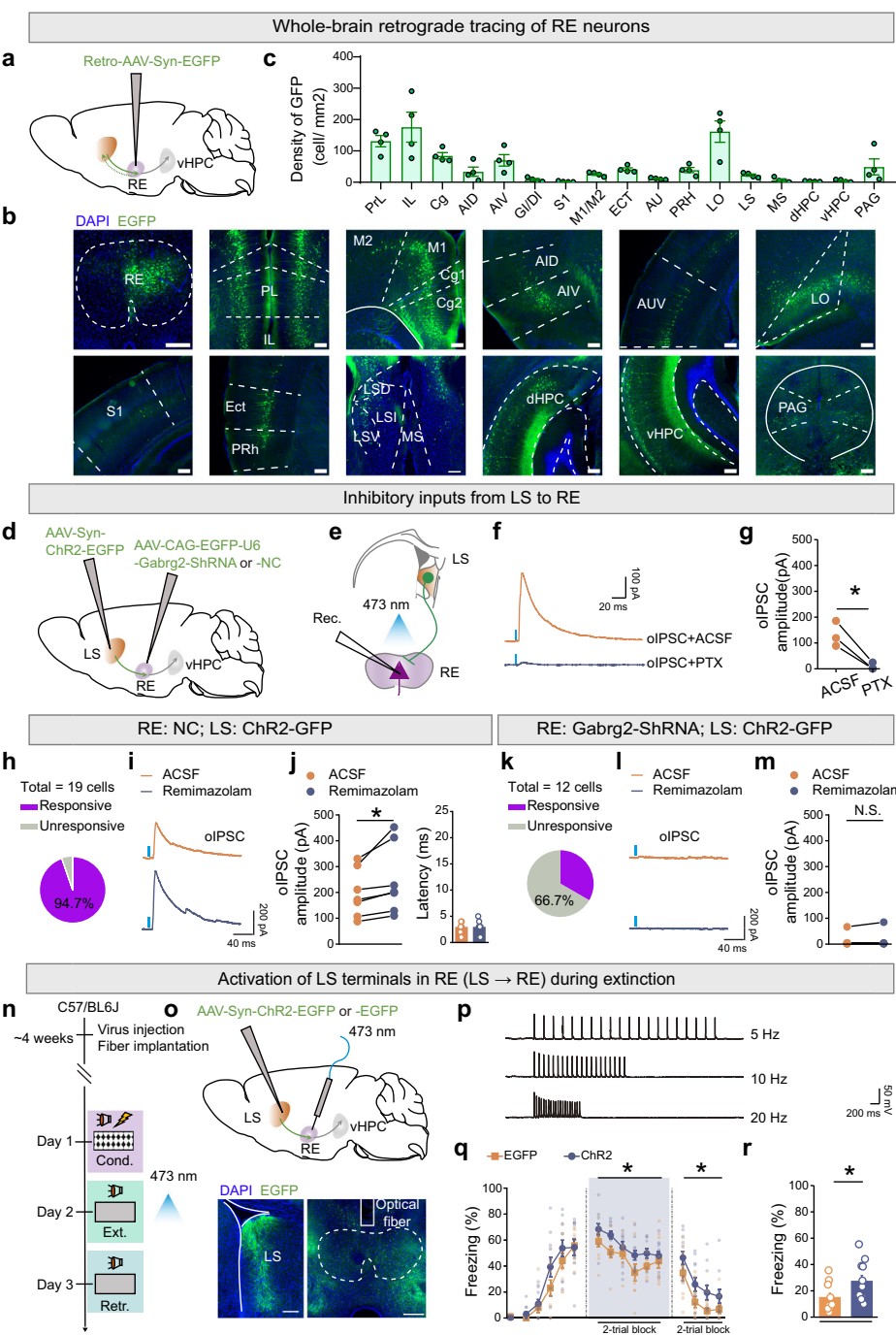

transsynaptic AAV expressing Cre (AAV1-Syn-Cre) into RE and Cre-dependent AAV expressing ChR2-mCherry (AAV-EF1α-DIO-ChR2-mCherry) into vHPC. Simultaneously, we injected retrograde AAV expressing mCherry (Retro-AAV-Syn-mCherry) into RE to visualize LS-RE projectors for further electrophysiological validation (Fig. 6a). Whole-cell recordings were then performed on LS-RE projectors while optogenetically activating vHPC axon terminals in LS. Both oEPSCs and disynaptic oIPSCs were observed in LS-RE projectors (Fig. 6b–e), with oEPSCs showing short onset latencies (Supplementary Fig. 7), indicative of monosynaptic excitatory inputs in hippocamposeptal circuits. In contrast, oIPSCs displayed significantly longer latencies (Supplementary Fig. 7) and were completely blocked by CNQX (20 µM) plus APV (50 µM), suggesting disynaptic inhibition. These findings support the idea that vHPC provides both direct excitation and indirect inhibition to LS-RE projectors likely through the

activation of local GABAergic interneurons and/or lateral inhibition from neighboring LS neurons (Fig. 6f). Together with the results shown above that systemic treatment of remimazolam prior to extinction learning significantly reduced vHPC activation but enhanced LS activation during extinction retrieval (Fig. 1d–f), we favored the notion that vHPC sends the inhibitory-dominant neural projections to LS for extinction control.

Behaviorally, optogenetic activation of the inhibitory-dominant projections from the vHPC to LS projections, which receive excitatory inputs from RE (Fig. 6g–i), presumably amplifies inhibition onto the LS → RE projections, characterized by GABAergic connectivity. This action disinhibits RE activation, ultimately enhancing fear extinction efficacy during both learning and retrieval phases (Fig. 6j, k). This suggests that the RE → vHPC → LS pathway promotes fear extinction, with the altered LS activity likely further influencing RE to regulate fear extinction.

**Fig. 6 | Activation of the RE → vHPC → LS pathway facilitates fear extinction, reversed by inhibiting RE-vHPC projectors. a** Schematics of AAV injections (*Upper*) and whole-cell patch clamp recording (*Lower*). Example traces of oIPSC (**b**) and oEPSC (**d**) from LS-RE projectors receiving vHPC inputs innervated by RE before and after bath application of CNQX (20 µM) and APV (50 µM). Statistics are as follows: two-tailed paired Student's *t*-test, **c** $t_{(3)} = 4.617$, *$P = 0.0191$; **e** $t_{(3)} = 4.078$, *$P = 0.0266$, n = 4 cells. **f** Schematic of the closed-loop model for fear extinction. **g, l** Experimental design. **h, m** Schematics of AAV injections. **i** Representative images of mCherry expression (red) in vHPC and optical fiber implantation in LS. **n** Representative images of mCherry expression (red) in RE. Scale bar, 200 µm. **o**, Example traces of stepping current injection induced action potentials treated before (*Upper*) and after (*Lower*) bath application of CNO (10 µM). **j, p**, Time course of freezing responses. **k, q** Freezing responses during extinction retrieval. **j, k** Effects of activating vHPC → LS projections receiving RE inputs on fear extinction. mCherry group, n = 6 mice; ChR2 mice, n = 7 mice. Statistics are as follows: **j**, two-way repeated measures ANOVA, Cond.: $F_{(1, 11)} = 2.168$, $P = 0.1690$; Ext.: $F_{(1, 11)} = 11.35$, **$P = 0.0063$; Retr.: $F_{(1, 11)} = 5.370$, *$P = 0.0408$. **k** two-tailed unpaired Student's *t*-test, $t_{(11)} = 2.317$, *$P = 0.0408$. **p, q** Effects of chemogenetic inactivating RE-vHPC projectors on optogenetic activating vHPC-LS projections receiving RE inputs on fear extinction. Vehicle group, n = 5 mice; CNO group, n = 5 mice. Statistics are as follows: **p** two-way repeated measures ANOVA, Cond.: $F_{(1, 8)} = 1.472$, $P = 0.2597$; Ext.: $F_{(1, 8)} = 52.29$, ***$P < 0.0001$; Retr.: $F_{(1, 8)} = 3.900$, $P = 0.0837$. **q** two-tailed unpaired Student's *t* test, $t_{(8)} = 1.975$, $P = 0.0837$. Data are presented as mean ± SEM.

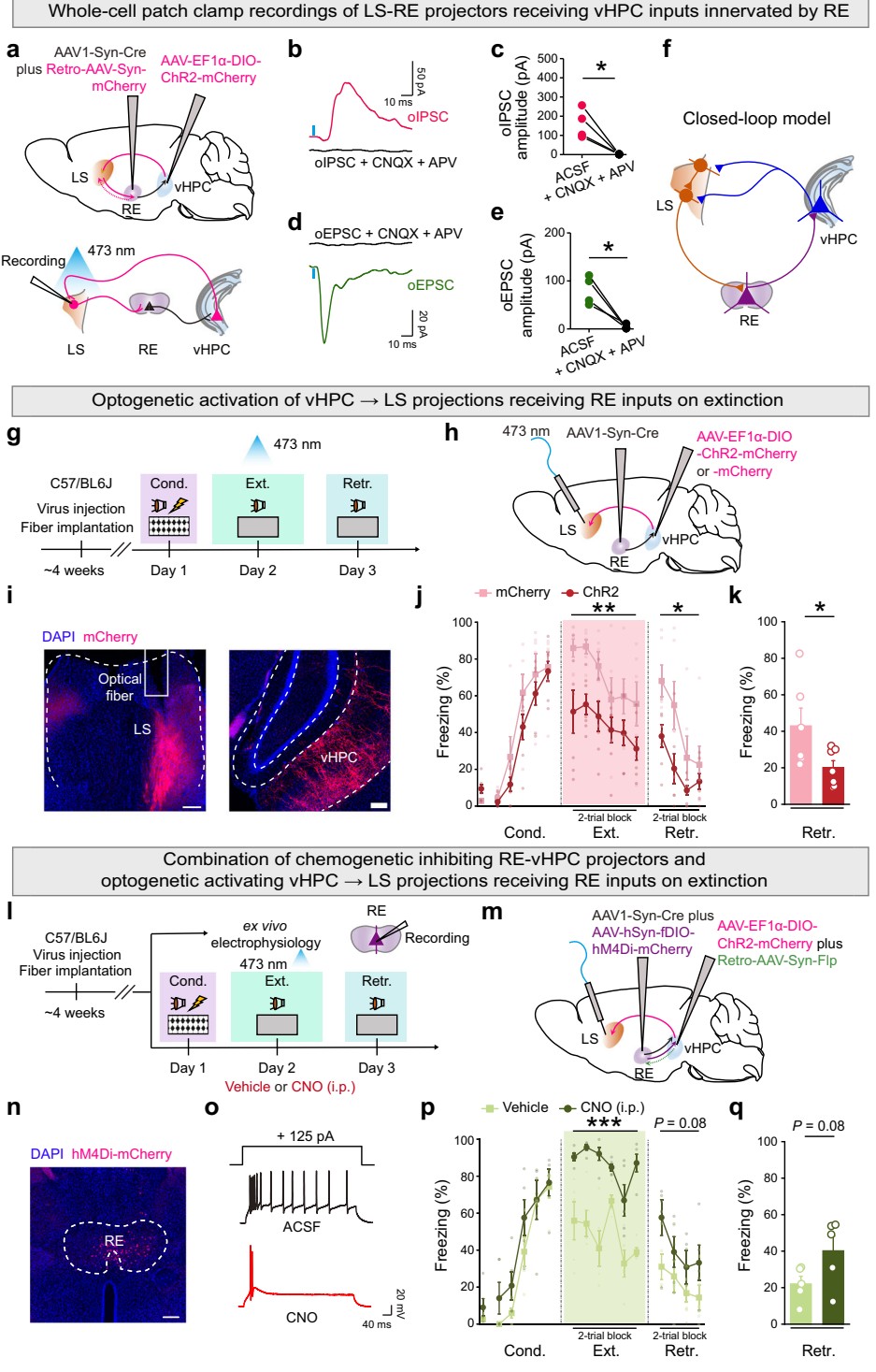

To establish the connection between the RE → vHPC → LS pathway and RE-vHPC activity during fear extinction, we utilized a similar manipulation as depicted in Fig. 6h, together with injection of retrograde AAV expressing Flp (Retro-AAV-Syn-Flp) into vHPC and Flp-dependent retrograde AAV expressing inhibitory hM4Di (AAV-EF1α-fDIO-hM4Di-mCherry) into RE (Fig. 6l–n). This enabled simultaneous chemogenetic inhibition of RE-vHPC projectors upon the administration of clozapine-N-oxide (CNO). Electrophysiologically, CNO significantly reduced neuronal firing in RE-vHPC projectors (Fig. 6o). Behaviorally, CNO administration (i.p.) at a dose of

1 mg/kg reversed the facilitation of fear extinction induced by optogenetic activation of the RE → vHPC → LS pathway (Fig. 6p, q). This observation supports the idea that RE-vHPC projectors serve both upstream and downstream functions in LS, contributing to the overall positive feedback circuit promoting fear extinction (Fig. 6f). Namely, RE directs a monosynaptic excitatory pathway to the vHPC, which, in turn, sends a feedforward inhibitory projection to GABAergic neurons in LS. This is coupled with long-range GABAergic projections from LS to RE, a region responsible for the fear extinction actions of remimazolam.

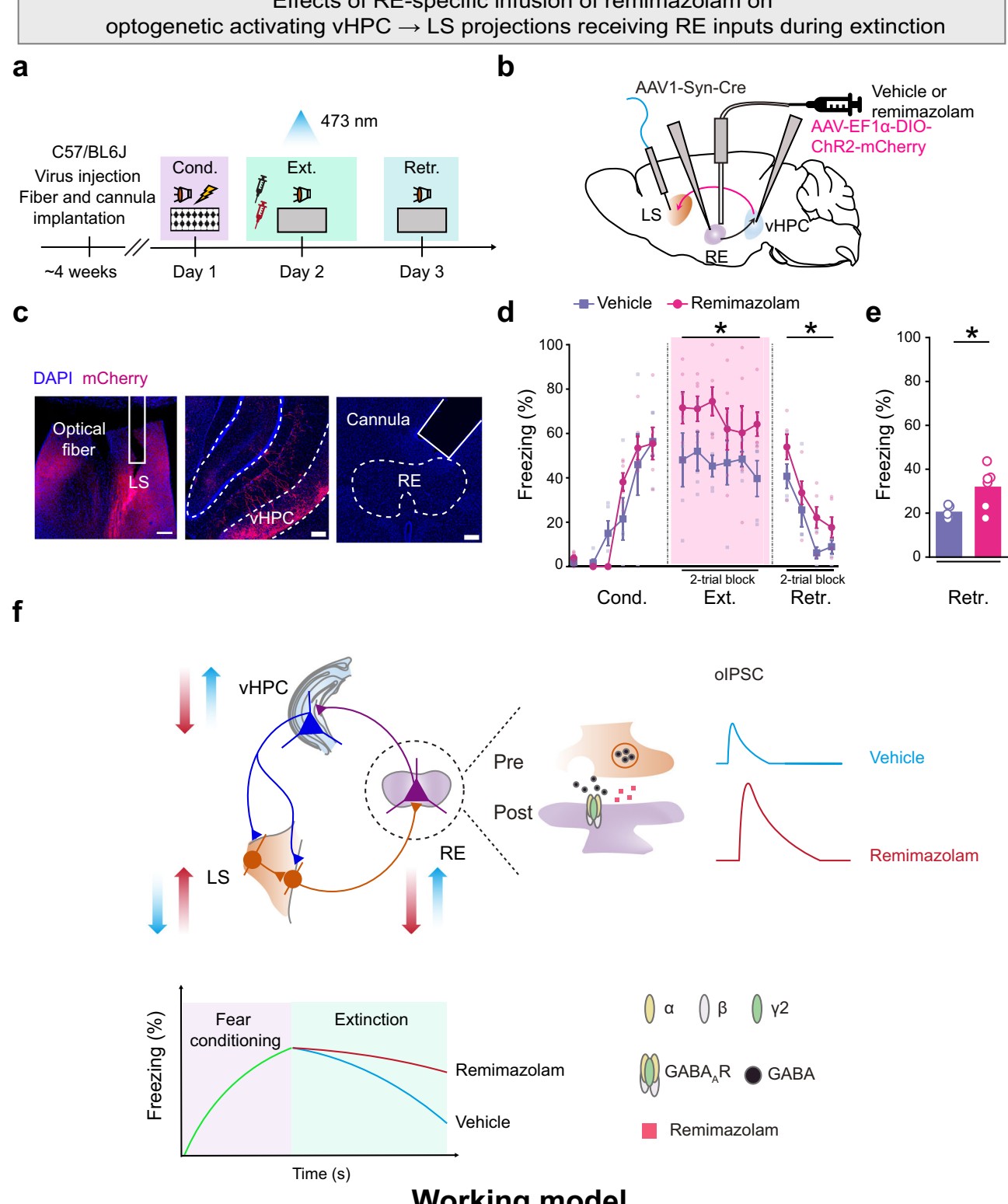

Effects of RE-specific infusion of remimazolam on optogenetic activating vHPC → LS projections receiving RE inputs during extinction

**Working model**

## RE-specific remimazolam nullifies the effects of activating RE → vHPC → LS pathway on fear extinction

Finally, we investigated whether remimazolam specifically targeting the RE would attenuate the activity of the RE → vHPC → LS pathway during fear extinction. Employing the same viral injections as described in Fig. 6h, we further placed a drug cannula above the RE to deliver remimazolam

(Fig. 7a–c). Notably, RE-specific remimazolam, akin to the chemogenetic inhibition of RE-vHPC projectors mentioned earlier (Fig. 6l–q), nullified the effects of optogenetically activating the RE → vHPC → LS pathway on fear extinction during both extinction learning and retrieval (Fig. 7d, e). Therefore, RE-specific remimazolam disrupts the overall positive feedback circuit construct of the RE → vHPC → LS pathway for fear extinction. This

**Fig. 7 | RE-specific remimazolam nullifies the effects of activating RE →
vHPC → LS pathway on fear extinction. a** Experimental design. **b** Schematics of
AAV injections. **c** Representative images of embedding sites for cannula and optical
fiber in LS (*Left*) and RE (*Right*) respectively, as well as mCherry expression (red) in
vHPC (*Middle*). Scale bar, 200 μm. **d**, **e** Effects of delivering remimazolam in RE
with activation of vHPC → LS projections receiving RE inputs on fear extinction.
Vehicle group, *n* = 5 mice; Remimazolam group, *n* = 6 mice. **d** Time course of
freezing responses to the CS. Statistics are as follows: two-way repeated measures
ANOVA, Cond.: $F_{(1, 9)} = 0.09548$, $P = 0.7644$; Ext.: $F_{(1, 9)} = 6.373$, $^*P = 0.0325$; Retr.:
$F_{(1, 9)} = 6.373$, $^*P = 0.0325$. **e** Freezing responses during extinction retrieval. Statistics

are as follows: two-tailed unpaired Student's *t*-test, $t_{(9)} = 2.550$, $^*P = 0.0312$.
**f** Schematic of the working model of remimazolam in RE that undermines fear
extinction through the modulation of hippocamposeptal circuits. RE-specific
remimazolam disrupts the overall positive feedback circuit construct of the RE →
vHPC → LS pathway for fear extinction. RE-specific remimazolam decreases the
excitability of the RE by potentiating the GABA$_A$R function on the postsynaptic
membrane of RE neurons. This, in turn, reduces the excitatory inputs from the RE to
the vHPC, which further reduces the feed-forward inhibitory inputs from the vHPC
to the LS, which increases the inhibitory inputs from LS to the RE. Data are presented
as mean ± SEM.

process involves remimazolam-induced reduction in RE excitability by
enhancing GABA$_A$R function on the postsynaptic membrane of RE neu-
rons. Consequently, this decreased RE excitability leads to a decline in
excitatory inputs from the RE to the vHPC, subsequently reducing the feed-
forward inhibitory inputs from the vHPC to the LS. Ultimately, this cascade
results in an increase in inhibitory inputs from LS to the RE. These circuit
mechanisms shed light on the inhibitory effects of remimazolam on fear
extinction (Fig. 7f). Targeting these mechanisms may offer a solution to the
challenge of combining benzodiazepine anxiolytics with extinction-based
exposure therapy.

## Discussion

Benzodiazepines have been used extensively as anxiolytics in the treatment
of anxiety and PTSD. However, benzodiazepine-induced disruption of
conditioned fear extinction, an important aspect of adaptation to environ-
mental change in the treatment of various mental disorders, poses a notable
challenge. The objective of the current study was to investigate the circuit
mechanisms underlying this issue by utilizing remimazolam, an ultra-short-
acting benzodiazepine that has been proposed to potentially circumvent the
combined effects of sedation, hypnosis, and anesthesia. Our comprehensive
assessment of neural circuits associated with fear extinction affected by
remimazolam revealed that it decreased RE and vHPC activation but
increased LS activation. Behaviorally, systemic, or RE-specific administra-
tion of remimazolam inhibited fear extinction. Notably, remimazolam
heightened GABAergic transmission and reduced neuronal excitability in
the RE. The potentiation effect of remimazolam on the long-range
GABAergic inhibition from the LS to the RE-vHPC projector were identi-
fied to be a key factor in elucidating the impaired fear extinction effect. We
further characterized that long-range excitatory projections from the vHPC
also indirectly engaged inhibitory neurons that counteract LS activity. This
in turn explained the contrasting effects of remimazolam-altered neuronal
activation on vHPC versus LS in the scenario of fear extinction. While
optogenetic activation of the RE → vHPC → LS pathway enhanced fear
extinction, chemogenetic inhibition of the RE-vHPC projector reversed this
effect, suggesting that the overall positive feedback loop created by the RE,
vHPC, and LS collectively drives fear extinction. Notably, remimazolam
targeting the RE disrupted this circuit, thereby nullifying the promotion of
fear extinction. This precise understanding of the circuit dynamics grants
valuable insight into the difficulties posed by the combination of anxiolytics,
such as remimazolam, and extinction-based exposure therapies.

The RE emerges as a central thalamic hub, orchestrating emotional and
cognitive functions across diverse brain states. Through its connections with
the mPFC and the hippocampus, the RE intricately facilitates the coupling of
cortical-hippocampal slow oscillatory activity, particularly vital for sleep-
related memory consolidation[37,50]. Moreover, the RE actively influences the
communication dynamics within the prefrontal-hippocampal pathway,
exerting its impact during both activated and deactivated states, reminiscent
of both rapid eye movement (REM) and non-REM sleep[51]. The multifaceted
involvement of the RE is emphasized by distinct firing patterns observed in
different populations of RE neurons during various sleep–wake states[52].
These different firing patterns determines the specificity and generalization
of memory attributes linked to a particular context. This occurs as the RE
processes information originating from the mPFC en route to the
hippocampus[32]. Additionally, the RE is indispensable for hippocampal-

dependent encoding of precise contextual memories, contributing to the
discrimination between safe and dangerous environments[53]. Further, the RE
to hippocampus pathway signals suppression, extinction, and discrimina-
tion of contextual fear memory[36]. In the cued auditory fear memory, the
mPFC-RE projection mediates fear extinction[34], dependent on the coordi-
nation of prefrontal-hippocampal synchrony to suppress extinguished
fear[43]. The RE to BLA pathway underlies the extinction of remote fear
memory[41]. Overall, the RE, with its multiple connections to the mPFC,
hippocampus, and BLA, actively engages in state-dependent transmission of
safety signaling, fine-tuning the level of fear memory. This functional
positioning of the RE and its related circuits resonates with previous
implications of benzodiazepines in counteracting safety signal recognition
and suppressing fear extinction[14].

In alignment with the roles of RE in behavioral state control, our study
unveiled that administering the ultra-short-acting benzodiazepine remi-
mazolam before extinction learning selectively dampened the activation of
the RE during extinction retrieval. This observation underscores the pivotal
role of the RE in mediating the benzodiazepine-induced control of fear
extinction. Building upon the reciprocal connections of RE with the hip-
pocampus and mPFC, we further discerned that remimazolam specifically
reduced the neuronal activation of the vHPC but not that of the mPFC
during extinction retrieval. These findings lend support to the idea that
GABA$_A$Rs in the RE are not only indispensable for remimazolam's reg-
ulatory impact on fear extinction but are also sufficient, particularly in RE-
vHPC projectors, to mediate the effects of remimazolam. This reinforces the
pivotal role of RE in steering hippocampal functions[54,55]. Given the pre-
valence of glutamatergic excitatory neurons in the RE[56], we favor such an
intriguing prospect: remimazolam might compromise RE → vHPC pro-
jections primarily through GABAergic mechanisms in the RE. However, we
acknowledge the possibility of GABAergic involvement in other brain
regions, which cannot be entirely ruled out.

As constituents of the hippocamposeptal circuits, the projection from
the vHPC to the LS orchestrates various adaptive behaviors[57], including
approach-avoidance conflict[58], feeding[59], and foraging-related memory[60].
Our study discerned that the vHPC, innervated by the RE, sends projections
characterized by feedforward inhibition to the LS, thereby exerting an
overall inhibitory control over LS activity through reciprocal connections
with RE neurons. This positive-loop circuit motif provides a cogent expla-
nation for our observation that systemic administration of remimazolam
results in the downregulation of RE and vHPC, accompanied by an upre-
gulation of LS activity during extinction retrieval. In light of this circuit
organization, remimazolam was found to electrophysiologically enhance
the long-range GABAergic transmission from LS to RE, contingent upon
the presence of GABA$_A$Rs on RE. Notably, different subtypes of LS neurons
are known to contribute to distinct facets of stress- or hedonic feeding-
related behaviors[61–67]. However, the specific subtypes of LS neurons
responsible for the remimazolam-mediated attenuation of fear extinction
behaviors await elucidation in future studies.

Remimazolam as a novel ultrashort-acting benzodiazepine possesses a
distinctive pharmacological profile that remains partially understood, par-
ticularly in vivo. While it demonstrates remarkable efficacy as an anesthetic
in clinical settings, individuals recovering from remimazolam-induced
anesthesia often experience robust amnesia even after consciousness is
restored[68]. However, there is a lack of consistent animal studies[69,70]

addressing the effects of remimazolam on general learning and memory, highlighting the need for further research to elucidate its intricate impacts on mnemonic processes. In light of the observed effects of remimazolam on fear extinction, we propose that potential mechanisms may involve a remimazolam-induced state characterized by the absence of retention of specific extinction memory. This hypothesis presents an intriguing avenue for exploration in future studies. Nonetheless, the findings of the present study underscore a noteworthy role for remimazolam in impeding fear extinction, adding complexity to its mechanisms beyond its original intended purpose for safe anesthesia.

In conclusion, our findings elucidate that remimazolam markedly impedes fear extinction by acting on GABA$_A$Rs in the RE, thereby influencing the intricate hippocamposeptal circuit. The identified positive feedback circuit construct involving RE-vHPC-LS-RE functions as a rheostat in fear extinction. Targeted disruption of this circuit, particularly when remimazolam is delivered specifically to the RE, provides detailed mechanistic insights into the complexities associated with benzodiazepine use in exposure therapy. This study underscores the comprehending on the neural circuits orchestrating fear extinction, laying a solid foundation for refining treatment strategies for anxiety disorders and PTSD.

## Methods

### Animals

We have complied with all relevant ethical regulations for animal use. The animal experiments strictly adhered to ethical guidelines established by the Animal Ethics Committee of Shanghai Jiao Tong University School of Medicine and the Institutional Animal Care and Use Committee (Department of Laboratory Animal Science, Shanghai Jiao Tong University School of Medicine), in accordance with policy DLAS-MP-ANIM.01-05. Conscious, unrestrained male mice aged 8 to 12 weeks, all of the C57BL/6J strain, were used in the studies. These mice were bred in a specific pathogen-free environment in the laboratory animal facilities, maintaining optimal conditions with temperatures ranging from 21 to 23 °C, humidity levels between 40% and 60%, and a consistent 12-h light/dark cycle. Standard rodent food and water were continuously available to the mice. Adult male mice aged 6 to 12 weeks were selected for various experimental procedures, conducted exclusively during the daylight period of the light/dark cycle, strictly following institutional guidelines.

It is important to acknowledge some limitations of our study. Fear extinction inherently exhibits high behavioral variability, influenced by numerous uncontrolled factors such as memory acquisition strength. Achieving consistent extinction levels throughout the study poses a challenge due to these inherent variations, compounded by constraints on the number of experimental animals used, leading to relatively larger variability in fear extinction control. While efforts were made to uphold internal validity by consistently involving the same cohort of animals in both control and experimental groups, fluctuations in freezing levels within the control groups may affect the interpretation of results.

### Fear conditioning and extinction

The entire auditory fear conditioning and extinction process were meticulously executed using the Ugo Basile Fear Conditioning System (UGO BASILE srl, Italy). Initially, mice underwent a thorough familiarization period with the conditioning chamber over three successive days. The chambers, measuring 17 cm × 17 cm × 25 cm, were equipped with stainless-steel shocking grids and integrated with a precision feedback current-regulated shocker from UGO BASILE srl, Italy.

During the fear conditioning phase on day 1 in context A, characterized by black-and-white checkered wallpapers, mice were individually exposed to five pure tones (CS; 4 kHz, 76 dB, 30 s each) at varied intervals (20–180 s), each synchronized with a foot shock (US; 0.5 mA, 2 s). The ANY-maze software (version 7.2, Stoelting Co., USA) ensured precise control over tone and shock delivery. Post-conditioning, mice were returned to their home cages after a 60-s interval, and the chamber was cleaned with 75% ethanol.

For the extinction learning phase on day 2 in context B, featuring a gray non-shocking plexiglass floor and dark gray wallpaper, mice previously exposed to context A underwent 12 CS sessions (4 kHz, 76 dB, 30 s each) without foot shocks. On day 3, during extinction retrieval, mice experienced 8 CS-only presentations in context B. The chamber, housed within a sound-attenuating enclosure, featured a ventilation fan and a singular house light from UGO BASILE srl, Italy.

Mouse activity within the chamber was recorded using a near-infrared camera and analyzed in real time by the ANY-maze software. The software calculated a "freezing score" based on a sophisticated algorithm that detected movement by comparing each pixel with its predecessors. A lack of detected movement, indicated by flickering pixels, suggested freezing. The software provided a frame-by-frame freezing score, automatically quantifying and analyzing freezing behavior, defined as a cessation of movement for more than 2 s. For animals with optical fibers, potential interference with light stimuli required independent scoring of freezing behavior by an experienced experimenter in a double-blind manner. The duration of freezing during each tone presentation was carefully measured and documented. While the freezing response during fear conditioning was calculated by the percent of freezing time CS during individual trials, those during extinction learning and retrieval were calculated by the average freezing responses of 2 consecutive trials (referred to as a 2-trial block).

### Open field test

The behavioral assessments were conducted in the designated experimental room. Mice, housed in their home cages, were transferred to the experimental room and acclimated for a minimum of 1 h prior to the experiments. To evaluate the effects of drugs on locomotor activity and basal anxiety, mice were administered systemic injections or intracerebral infusion were placed in the center of a square Plexiglas open field apparatus (40 × 40 × 35 cm) 15 min post-injection. They were then allowed to freely explore for a duration of 30 min. To investigate the impact of optogenetic activation of LS → RE projections on locomotor activity and basal anxiety, mice underwent an 18-min testing session, consisting of three rounds of 3-min period of light-off followed by 3-min period of light-on. Blue light (473 nm, 4–6 mW, 10 ms per pulse, 20 Hz) was unilaterally delivered targeted above the RE during the light-on phase. Between each test, the open-field arena was thoroughly cleaned with 70% ethanol. The ANY-maze software (UGO BASILE srl, Italy) was utilized to quantify the total distance traveled and the time spent in the center zone by the mice.

### Intraperitoneal injection procedure prior to behavioral testing

Approximately 15 min before the fear extinction or open field test, each mouse received an i.p. injection of remimazolam at doses of 2, 4, or 8 mg in 5 ml per kg body weight of mice. To pharmacologically counteract the effects of RE-specific remimazolam delivery, each mouse was given an i.p. injection of flumazenil at a dose of 4 mg in 5 ml per kg body weight of mice, administered 15 min before the remimazolam delivery. For the vehicle control, saline (0.9% NaCl, w/v) was injected instead. Following the injection, mice were housed in their home cages until the commencement of the behavioral tests.

### Viral constructs

The following viral vectors were procured from SunBio Biomedical Technology Co. Ltd. (Shanghai, China) for the current study: AAV-CAG-EGFP-Gabrg2-ShRNA (Serotype 2/9), AAV-CAG-EGFP-NC (Serotype 2/9), and AAV-U6-Gabrg2-ShRNA-DIO-Gabrg2*-2A-EGFP (Serotype 2/9). Additionally, AAV1-Syn-Cre (Serotype 2/1), Retro-AAV-Syn-Cre-mCherry (Serotype 2/retro), Retro-AAV-Syn-mCherry (Serotype 2/retro), AAV-Syn-ChR2-E123T-T159C-EGFP (Serotype 2/9), AAV-Syn-EGFP (Serotype 2/9), AAV-EF1α-DIO-ChR2-E123T-T159C-mCherry (Serotype 2/9), AAV-EF1α-DIO-mCherry (Serotype 2/9), Retro-AAV-Syn-Flp (Serotype 2/retro), AAV-EF1α-DIO-hM3Dq-mCherry (Serotype 2/9), AAV-EF1α-DIO-mCherry (Serotype 2/9), AAV-fDIO-hM4Di-mCherry (Serotype 2/9) were packaged by BrainVTA Co. Ltd (Wuhan,

China). All viral vectors were stored in aliquots at $-80\,°C$ until further use. Unless otherwise specified, the viral titers of AAVs for injection exceeded 10^12 viral particles per ml.

## Stereotaxic surgery

Mice, aged precisely between 6 and 7 weeks, underwent gentle sedation with a precisely measured dose of 1% (w/v) sodium pentobarbital administered via a single i.p. injection (10 ml per kg body weight of mice). Subsequently, each mouse was securely positioned in a state-of-the-art stereotactic frame with non-rupture ear bars (RWD Life Science, Shenzhen, China) to minimize discomfort. A midline scalp incision was carefully made, leading to the creation of symmetrical craniotomies using a microdrill with ultrafine 0.5-mm burrs. Specialized glass pipettes (tip diameter: 10–20 µm) for AAV microinjections were expertly crafted using a P-97 Micropipette Puller (Sutter glass pipettes, Sutter Instrument Company, USA). These micro-injection pipettes, initially filled with silicone oil, were connected to a precise microinjector pump (RWD Life Science, Shenzhen, China) to achieve complete air exclusion. AAV-containing solutions were injected at precise coordinates (anteroposterior to bregma, AP; lateral to the midline, ML; below the bregma, DV; in mm): RE: AP $-0.58$, ML $-0.1$, DV $-4.25$; LS: AP $+0.7$, ML $\pm0.4$, DV $-3.2$; vHPC: AP $-3.14$, ML $\pm3.16$, DV $-4.0$. Virus-containing solutions were injected into LS (0.2 µl/side, bilaterally), RE (0.15 µl/side, with a single central injection allowing bilateral infusion), vHPC (0.4 µl/side, bilaterally) at a rate of 0.1 µl/min. After injection, the pipette was maintained in position for an additional 10 min to ensure thorough diffusion of the injected substance within the target area. Mice were allowed a minimum recovery period of 4 weeks before subsequent behavioral and other assessments. At the experiment's conclusion, injection sites were meticulously examined to evaluate the expression of fluorescent proteins (EGFP or mCherry), providing crucial insights into injection precision and efficacy.

In optogenetic experiments, unilateral ceramic fiber optic cannulas (200 µm in diameter, 0.37 numerical aperture (NA), Hangzhou Newdoon Technology, China) were surgically implanted above the LS (coordinates: AP $+0.7$, ML $+0.4$, DV $-3$) and RE (coordinates: AP $-0.58$, ML $+0.1$, DV $-4.0$). These cannulas were firmly secured in position using acrylic dental cement and anchored with skull screws.

For intracranial injection experiments, infusion cannulas (500 µm in diameter, 5 mm in length, RWD Life Science, Shenzhen, China) were carefully positioned above the RE (coordinates: AP $-0.58$, ML $-2.46$, DV $-4.9$, 30° angle, with a single central injection allowing bilateral infusion). The infusion cannulas were affixed in place using acrylic dental cement and anchored with skull screws. The infusion cannulas cap was put on after the operation with a length of 5.5 mm to prevent infection, and the experiment could be performed after two weeks of recovery.

## Optogenetic manipulations during fear extinction learning

In the course of behavioral experiments, a blue LED light source (473-nm wavelength, Hangzhou Newdoon Technology Co. Ltd) was utilized for photostimulation. This light source was connected to a flexible patch cord equipped with connectors at both ends for convenient attachment. The optical fiber implanted in the mouse (200 µm in diameter, 0.37 NA) was linked to the optic patch cord through ceramic mating sleeves. Blue light (473 nm, 4–6 mW) was delivered in 10 ms pulses at 20 Hz during the presentation of each 30-s CS, with the light delivery covering the CS exposure for the extinction and renewal test (extending 5 s before and after the CS).

## Intracranial drug infusions prior to fear extinction learning

Intracranial drug infusions during fear extinction learning were conducted 15 min before the first day of extinction. During the infusion process, mice were briefly head-restrained, and stainless-steel obturators were removed to allow the insertion of injection cannulas into the guide cannulas. The injection cannulas protruded 0.5 mm from the tips of the

guide cannulas. The microinjection pipettes were filled with silicone oil to exclude air, then connected to the microinjector pump (Chorny, model: ZS100, China). Either vehicle (ACSF) alone or remimazolam (100 µM in ACSF, 0.5 µl) was injected into the RE at a rate of 0.5 µl/min, and the injector was left in place for 10 min to ensure adequate diffusion of the injectant.

## Slice electrophysiology

Whole-cell recordings were performed on brain slices from mice injected with AAVs, as illustrated in various figures. The preparation of these slices for electrophysiological studies was conducted by a researcher aware of the experimental group's identity. Meanwhile, electrophysiological data collection and analysis were independently carried out by a skilled experimenter, who was blinded to group assignments. Mice underwent deep anesthesia using 1% sodium pentobarbital before decapitation. Their brains were rapidly excised and immersed in oxygen-rich (95% $O_2$/5% $CO_2$) ice-cold ACSF. The ACSF composition included 125 mM NaCl, 2.5 mM KCl, 12.5 mM D-glucose, 1 mM $MgCl_2$, 2 mM $CaCl_2$, 1.25 mM $NaH_2PO_4$, and 25 mM $NaHCO_3$, with a pH between 7.35 and 7.45. Coronal slices (300 µm thick) from specific brain regions (RE, LS, or vHPC) were sliced using a Leica VT1200S vibratome. After a 40-min recovery in oxygenated ACSF at ~34 °C, slices were placed in a recording chamber and continuously perfused with oxygenated ACSF at a flow rate of 1–2 ml/min. Neurons in the RE, LS, or vHPC regions were targeted for patching, guided visually through an infrared differential-interference contrast microscope (BX51WI, Olympus, Japan) and an optiMOS camera (QImaging, Teledyne Imaging Group, USA). During electrophysiological experiments, slices were maintained in well-oxygenated ACSF at around 34 °C. Recordings were made using an Axon 200B amplifier (Molecular Devices, USA), with membrane currents sampled and analyzed through a Digidata 1550 interface and software (Clampex and Clampfit, pCLAMP 10.5, Molecular Devices, USA). Cells with access resistance between 15–30 MΩ and less than 20% change were included in analyses. Photostimulation of ChR2-expressing neurons involved a collimated LED (Lumen Dynamics Group Inc, USA) at a 473 nm wavelength, connected to the Axon 200B amplifier. The brain slice received illumination via a 40× water-immersion lens (LUMPLFLN 40XW, Olympus, Japan), with photostimulation intensity (2–18 mW/mm$^2$) and duration managed by the Digidata 1550 and pClamp 10.5 software. The effectiveness of the ChR2-expressing virus was verified by assessing the number of action potentials (APs) elicited at various blue-light stimulation frequencies (1 ms; 5, 10, and 20 Hz).

## Optically evoked IPSCs and EPSCs

Optically evoked IPSCs (oIPSCs) and EPSCs (oEPSCs) were induced to elicit synaptic responses in the RE or LS by optogenetic photostimulation of LS axons or those in LS-RE projectors. Similarly, vHPC axons or axons of RE-vHPC-LS projectors were stimulated. This was accomplished by illuminating each brain slice every 20 s with 5-ms blue-light pulses. To ensure that only monosynaptic activities were recorded in the EPSC experiments, photostimulation intensities were meticulously adjusted to achieve 30–50% of the maximum synaptic response. Optical stimulation of ChR2-expressing axons utilized a 473 nm peak wavelength blue collimated LED (Lumen Dynamics Group Inc, USA), connected to an Axon 200B amplifier. The brain slice in the recording chamber was illuminated through a 40× water-immersion objective lens (Olympus LUMPLFLN 40XW, Japan), with the intensity and duration of photostimulation regulated by the stimulator and the Digidata 1550 and pClamp 10.5 software, respectively.

For recording oEPSCs or oIPSCs, recording pipettes (3–5 MΩ) were filled with a solution comprising 132.5 mM Cs-gluconate, 17.5 mM CsCl, 2 mM $MgCl_2$, 0.5 mM EGTA, 10 mM HEPES, 2 mM $Na_3$-ATP, and 5 mM QX-314. The solution's pH was adjusted to 7.3 using CsOH, with an osmolarity between 280 and 290 mOsm. To assess the impact of remimazolam on oIPSCs and oEPSCs, 100 µM of remimazolam was added to the ACSF following the recording of oIPSCs and oEPSCs from the same cell in

standard ACSF. During these recordings, the patched RE neurons were voltage-clamped at 0 mV for oIPSCs and at −70 mV for oEPSCs.

## Electrically evoked IPSCs and EPSCs

Electrically evoked IPSCs (eIPSCs) and EPSCs (eEPSCs) were recorded from the RE neurons. Patch pipettes were filled with a $Cs^+$-based solution containing (in mM): 132.5 Cs-gluconate, 17.5 CsCl, 2 $MgCl_2$, 0.5 EGTA, 10 HEPES, 2 $Na_3$-ATP, 5 QX-314, with pH adjusted to 7.3 using CsOH, and osmolarity maintained at 280–290 mOsm. Stimulation was delivered through an ISO-Flex stimulus isolator (A.M.P.I.), with the stimulating electrode placed near RE, approximately 200–500 mm away from the recorded cell, to induce eIPSCs. To assess the effects of remimazolam on eIPSCs and eEPSCs, remimazolam (100 μM) was added to the ACSF after recording eIPSCs and eEPSCs from the same cell in ACSF. The patched RE neurons were voltage-clamped at 0 mV when recording eIPSCs and voltage-clamped at −70 mV when recording eEPSCs.

## Spiking firing

Spike firing activity and membrane properties of various populations of different RE, LS, or vHPC neurons were assessed using an internal solution comprising (in mM): 145 potassium gluconate, 5 NaCl, 10 HEPES, 2 MgATP, 0.1 $Na_3$GTP, 0.2 EGTA, and 1 $MgCl_2$ (280–300 mOsm, pH 7.2 with KOH). The data were analyzed using the MiniAnalysis Program (Version 6.0.1, Synaptosoft, USA) with an amplitude threshold set at 20 mV.

## Histology and fluorescent immunostaining

Mice were deeply anesthetized with 1% (w/v) sodium pentobarbital (i.p.) and subsequently underwent transcardial perfusion with saline followed by ice-cold phosphate-buffered saline (PBS). The brains were then extracted and fixed in a 4% paraformaldehyde solution in PBS. Coronal brain sections encompassing the entire LS, RE, and vHPC, were cut into slices of either 30-μm or 50-μm thickness using a VT1200S vibratome (Leica, Japan). These slices were prepared for subsequent analysis to assess the efficiency and specificity of viral infections. Subsequently, sections were incubated for 15 min in a 4,6-Diamidino-2-phenylindole dihydrochloride hydrate (DAPI) solution (1:5000; catalog no. D1306, ThermoFisher Scientific, USA), followed by four 15-min washes in PBS with 0.1% Tween-20. Prepared slides were covered with glass coverslips in darkness using mounting media and readied for microscopic examination.

For c-fos immunofluorescent staining, brain slices were initially blocked in a permeable buffer (0.3% Triton X-100 in PBS) with 10% donkey serum for 1 h at room temperature. Subsequently, slices were incubated overnight at 4 °C with rabbit anti-c-fos primary antibody (1:500; Cell Signaling Technology, catalog no. 2250, USA) in a permeable buffer containing 2% donkey serum. Following this, the slices underwent four 15-min washes in PBST (0.1% Tween-20 in PBS). They were then treated with donkey anti-rabbit IgG (H + L) Alexa Fluor secondary antibodies (1:200; Alexa Fluor™ 555, catalog no. A31572, ThermoFisher Scientific, USA) and DAPI (1:5000; catalog no. D1306, ThermoFisher Scientific, USA) in PBS buffer for 2 h at room temperature. After four washes in PBS-T, the slices were mounted on glass slides using mounting media. Fluorescent markers were imaged using Olympus confocal microscopes (VS120 or VS200, Japan). ImageJ software (NIH Image, version 1.8.0, USA) facilitated the quantification of immunofluorescent labeling through manual counting of fluorescence-positive cells. Standard histological methods and confocal microscopy were employed to confirm the locations of optical fiber tips for all behavioral experiments.

For c-fos quantification, brain slices were collected at intervals of 240 μm. The stained slices were then aligned with the corresponding coronal section of Allen Mouse Brain Atlas for statistical analysis. At least 2 brain slices were collected from each brain region of every mouse for analysis. Specially for the RE, 3 slices were taken from each mouse, with registered brain map coordinates across the anterior-posterior (AP) axis noted as AP −0.58, −0.82, and −1.06. The ImageJ software (NIH Image,

version 1.8.0, USA) was employed for the identification of c-fos-positive cells and subsequent statistical analysis utilizing the analyze particles method.

For the immunofluorescent staining against the $GABA_AR$ γ2 subunit, brain was sectioned into 50 μm slices after fixation with a 9% glyoxal/8% acetic acid fixative, employing an improved approach to address immunohistochemical challenges associated with molecules situated within specialized neuronal components[71]. These brain slices were initially incubated in a permeable buffer (0.1% Triton X-100 in PBS) with 10% donkey serum for 1 h at room temperature to block nonspecific binding sites. Subsequently, the slices were incubated overnight at 4 °C with mouse Anti-$GABA_AR$ γ2 primary antibody (1:500; catalog no. ab307231, Abcam, United Kingdom) in a permeable buffer containing 2% donkey serum. Following this, the slices underwent four 15-min washes in PBST (0.1% Tween-20 in PBS). They were then treated with donkey anti-mouse IgG (H + L) Alexa Fluor secondary antibodies (1:200; Alexa Fluor™ 647, catalog no. A31571, ThermoFisher Scientific, USA) and DAPI (1:5000; catalog no. D1306, ThermoFisher Scientific, USA) in PBS buffer for 2 h at room temperature. After four washes in PBS-T, the slices were mounted on glass slides using mounting media. Fluorescence was imaged using Olympus confocal microscopes (VS120 or VS200, Japan), and ImageJ software (NIH Image, version 1.8.0, USA) was employed for quantification.

## Statistics and reproducibility

This study draws data from a minimum of three independent biological replicates, ensuring consistent replication across diverse animal cohorts from distinct litters. Each micrograph was iterated at least three times, with all replication attempts yielding successful outcomes. Graphical representations were meticulously generated using Origin Software (version 9.5, OriginLab Corporation, Northampton, MA, USA). Data are primarily presented as mean ± SEM, with individual data points featured prominently in histograms, representing specific values and sample counts for each experimental condition. Prior to analysis, data distributions were rigorously tested for normality, and Levene's test assessed the homogeneity of variance among groups. Statistical comparisons utilized various methods, such as the two-tailed Student's $t$ test, one-way analyses of variance (ANOVAs), and two-way repeated measures ANOVAs. Post hoc analyses applied Bonferroni's corrections for multiple comparisons. Sophisticated tools like GraphPad Prism (version 8.0.2, GraphPad Software, Inc., USA) and Office 2019 (Microsoft, USA) facilitated statistical analyses. A significance level of $P < 0.05$ was adopted, denoted as $^*P < 0.05$, $^{**}P < 0.01$, and $^{***}P < 0.001$ in the results. In cases of multiple comparisons, distinct markers ($^\#P < 0.05$, $^{\#\#}P < 0.01$, and $^{\#\#\#}P < 0.001$) were used. Results lacking statistical significance are labeled as N.S. (non-significant).

## Reporting summary

Further information on research design is available in the Nature Portfolio Reporting Summary linked to this article.

## Data availability

All data needed to evaluate the conclusions of the present study are present in the main paper and/or the Supplementary Information. The source data behind the graphs in the paper can be found in Supplementary Data 1. Additional data are available from the corresponding authors upon request.

## Code availability

All the custom code used in the study is available from the corresponding authors on reasonable request.

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

## Acknowledgements

We thank the entire Tian-Le Xu and Wei-Guang Li labs for advice, technical support, and discussion. The present study was supported by grants from the STI2030-Major Projects (2021ZD0202800), the National Natural Science Foundation of China (31930050, 82371289, 32071023, 32371078, and T2293734), the Program of Shanghai Academic/Technology Research Leader (22XD1420700), the Shanghai Municipal Health Commission (2022XD046), the Shanghai Rehabilitation Medicine Association (2023JGKT29), and Innovative Research Team of High-Level Local Universities in Shanghai.

## Author contributions

H.C., T.-L.X., Y.L., M.-Z.Z., and W.-G.L. conceptualized the study. H.C, T.-Z.Y., X.Y., Y.-J.W., Q.W., X.G., M.X., M.C., W.W., X.-N.L., and Y.-X.L. performed experimental research and data analysis. Y.S. and J.Z. contributed to data interpretation and experimental design. H.C., T.-Z.Y., X.Y., Y.L., M.-Z.Z., and W.-G.L. wrote the manuscript with contributions from all authors. All authors read and approved the final manuscript.

## Competing interests

The authors declare no competing interests.
