## [Peer Review File · Communications Biology]

Reviewers' comments:

Reviewer #1 (Remarks to the Author):

In this paper, the authors investigate the impact of remimazolam, a short-acting benzodiazepine, on fear extinction learning and retrieval, as well as a potential circuit mechanism mediating its effects. First, they observed that systemic remimazolam hindered extinction learning and identified potential brain regions involved through brain-wide cFos screening, including the vHPC, LS, and thalamic RE. Focusing on the RE, the authors find that RE in part mediates remimazolam's effects, with GABA-A receptors in RE being required for its effects. They then relate their cFos quantification findings in the context of an RE-vHPC-LS positive feedback circuit with extensive use of anterograde and retrograde viral vector methods. They find that vHPC-projecting RE neurons require intact GABA-A signaling for proper extinction learning, and restoration recovers the effect of remimazolam on RE. Inhibition of RE from the LS mimics the effects of remimazolam. Connecting these two projections is the vHPC-LS projection, which was found to be largely inhibitory, with activation of it facilitating extinction learning and driving a "positive feedback loop." Overall, the manuscript adds to our understanding of benzodiazepine action and the role of RE circuitry in fear conditioning and extinction. The data is well-presented, and the text is largely written well. The following lists my concerns and suggestions for the authors.

Major Comments:

1. Knockdown of *gabag2* leads to remimazolam having no effect on RE. The data seem to also suggest that *gabag2* knockdown in RE results in a similar phenotype on extinction learning as remimazolam administration. One would expect that without reliable inhibition from LS due to nonfunctional GABA receptors in this feedback loop, RE neurons would be more activated/excitable. However, Fig. 3f vs 3fk in the ACSF condition suggest otherwise. Similarly, Fig. 3m vs 3o suggests that RE is "more" inhibited in vehicle-injected mice. Are RE neurons less excitable following gamma-2 knockdown? How do the authors interpret this? Please clarify this.
2. Are there any effects on learning and memory more generally due to remimazolam? Add discussion if there are literature.
3. That the RE sends excitatory projections to vHPC (line 325-326) is not clear until readers reach the end of the results. The authors may highlight evidence, either from their data or the literature, that show RE-HPC projections are excitatory (or that remimazolam-affected RE neurons are glutamatergic). It would be helpful to include this information earlier on, so that understanding the nature of this circuit is easier.
4. The methods should be checked to ensure that all information regarding performed experiments is included. They should also be in-depth enough to allow for reproducibility.
 - a. Details regarding systemic administration of remimazolam are not present.
 - b. More information related to trials vs blocks in extinction learning should be added. For example, "2-trial block" is not defined or described other than in the figures. Are pairwise comparisons (e.g. as in Fib 1b) between 2-trial-averaged blocks?
 - c. What is the rationale for the remimazolam concentrations used (50 μ M in vivo and 100 for electrophysiology)? Any previous literature?
 - d. Was the whole RE nucleus used for cFos quantifications? One slice? More details are needed.

Does the RE uniformly respond to remimazolam across the anterior-posterior axis?

5. Fig. 5f,g does not support the claim made in the sentence starting on line 259. A direct comparison between the two conditions mentioned might be helpful.

Minor Comments:

1. Some word choices seem out of place.

a. Intermediation as mentioned in the title and abstract could be better worded.

b. "Originates" (line 83) could be better put (sends?)

c. "Injected" (line 142) would be better put as infused

d. "Chemical genetic" (line 214) should be chemogenetic

2. Some phrases/sentences are unclear or undefined.

a. The sentence beginning on line 65 is difficult to understand.

b. Define oIPSCs (line 259)

c. The sentence beginning on line 307 should be stated clearer, as it could be a great summary of circuit and its functions as investigated up to that point.

3. The RE appears to encapsulate the xiphoid nucleus as it's drawn on Fig. 1e. Ensure that the xiphoid was not used for quantifications and update the ROI on 1e.

4. It would be helpful to mention in the main text (or show in the figure) that remimazolam was administered systemically/intraperitoneally for Figures 3-5.

5. EGFP expression for the reversible knockdown as shown in Fig. 4b does not give the impression that the virus expressed. An image comparable to the representative image in Fig. 3b, or a more representative one in general, should be used.

6. Fig. 6n's "hM4Di-mCherry" should be red, not green.

Reviewer #2 (Remarks to the Author):

Cheung et al. found that remimazolam potentiated GABA transmission in the RE and suppressed RE neuronal activity, resulting in the alteration of hippocamposeptal circuit function. This mechanism underlies how remimazolam hinders fear extinction. The study is intriguing and holds significant implications for comprehending the functional role and mechanism of remimazolam in fear extinction regulation. However, there is room for improvement in the writing as the logic in the introduction is not very clear. Additionally, some sentences lack clarity in the description. I have a few points listed as below.

1. Regarding the entire set of figures concerning freezing levels in the Ext and Retr phases, it is evident that the freezing levels for the vehicle varied widely across different groups (ranging from 20% to 50%). In contrast, the Remimazolam group exhibited a relatively stable freezing level, consistently at around 40-50% during the extinction retrieval session (except for fig1j, nearly 60% for remimazolam group). This discrepancy highlights a notable difference between the vehicle and Remimazolam, primarily attributed to the unstable freezing levels observed in the vehicle group.

The authors are encouraged to discuss these results and provide further explanations.

2. Why did the authors choose to use remimazolam at a dosage of 3.75 mg/kg, and how does this relate to its clinical application?
3. In Fig1b, the freezing levels observed for remimazolam during the extinction phase are even higher than the last two points seen during the conditioning phase. What could explain this finding? Additionally, if remimazolam administered via i.p. injection affects freezing levels in naïve mice?
4. What does “two-trial block” mean? Please include the description.
5. Line 133: dorsal part (LSR) should be LSD.
6. Fig1h, Line142: the images show that only one tract of the cannula above the RE, while the text reads that drug was bilaterally injected into the RE. Any typo?
7. Line 143: the use of “on the following day” is inappropriate, it is still referring to the same day.
8. Why did authors use 50 μ M Remimazolam for in vivo behavioral study, while use 100 μ M Remimazolam for slice recordings?
9. Fig2b and 2e, Additional sample traces with GABAAR antagonist or glutamate receptor antagonists should be included to confirm that evoked currents are IPSCs or EPSCs.
10. Fig2i,3f,3k: Rheobase (mA?), it should be pA.
11. Fig4b, it is hard to tell the EGFP expression in RE neurons. Please also include a high magnification image to allow the reader to see the expression of EGFP and mCherry in the RE.
12. Line1154, please verify the concentration of CNO used for ephys. 100 μ M maybe too high!
13. Line238-240, fig5b,c: “RE neurons received substantial inputs from the midbrain periaqueductal gray (PAG), basolateral amygdala (BLA), and nucleus accumbens (NAc)”. no images or data show BLA and NAc.
14. Fig5f,i,l: Please include the light application bar aligned with the current trace. What is the latency for oIPSC?
15. Fig5O, please also include a high magnification image to show the expression of GFP fibers are seen in the RE.
16. Please provide clarification in the methods section regarding whether the delivery of the virus or drug to the brain region is bilateral or unilateral, as well as details regarding the cannula implantation.
17. Fig5q: the labels are marked wrong (vehicle and Remimazolam?)
18. Fig6i, Fig7c, please include a high magnification image to show the expression of mCherry in the LS and RE.

Reviewer #3 (Remarks to the Author):

In the submission "An ultra-short-acting benzodiazepine in thalamic nucleus reuniens undermines fear extinction via intermediation of hippocamposeptal circuits" the authors address an interesting question, what is the neuronal mechanisms underlying the effect of remimazolam on the learning of conditioned fear extinction? The authors employ appropriate methodologies to answer the question and the experiments are well executed. However, the authors tend to overstate some results and omit some controls and discussion about important aspects of the work. This study has to be revised but definitely deserves publication.

Please find below my comments for each part/figure.

First of all, concerning the title of the paper. Why do the authors emphasize the fact that remimazolam is an ultra-short-acting benzodiazepine? They are not showing the dynamic of action of the remimazolam.

1/ The authors aim at characterizing the effect of remimazolam on the learning of conditioned fear extinction. Intraperitoneal and intracerebral injections of remimazolam before extinction have two effects: increasing freezing during extinction in presence of the drug (day2) and during retrieval in absence of the drug (day3). As stated in the title, abstract and text of the paper the authors conclude from these results that remimazolam undermines fear extinction. However, remimazolam does more than that. Vehicle and remimazolam injected mice show different levels of freezing during the first CS presentations of extinction showing an effect of remimazolam on fear expression itself. This result is not interpreted or discussed by the authors. Data in figure 1 panels c and j are the same. Please provide the right results.

2/ Taken together, the results of the c-fos (figure1f) and the patch recordings of RE neurons (figure2), show that remimazolam decreases neuronal excitability in RE neurons probably by the potentiation of GABAergic transmission. The authors use different values for the axis showing rheobase, eIPSC amplitude and neuron excitability throughout the manuscript. This does not help to compare the results between experiments.

3/ The authors aim at demonstrating that remimazolam potentiates GABAergic transmission in RE. For that they use a shRNA strategy to reduce the expression of the $\gamma 2$ subunit of GABA_A receptors. The authors mention in the text that the used viral strategy is well-verified. They should provide at least histological proof of the efficacy of this virus. In figure 3, the authors compare the effect of remimazolam in presence or absence of $\gamma 2$ subunit of GABA_A receptors within RE. The results are clear, it would be great to have the same axis values for all panels. Regarding the behavioral effect of such manipulations, the authors provide statistical comparisons of freezing levels only between vehicle and remimazolam for each viral condition. By looking at the data, it seems that AAV-CAG-EGFP-U6-gabag2-ShRNA-injected and AAV-CAG-EGFP-U6-NC-injected animals exhibit different levels of freezing when comparing vehicle conditions. If it's the case, it means that reducing the expression of the $\gamma 2$ subunit of GABA_A receptors in RE neurons affects fear expression and extinction learning. This result argues in favor of the authors hypothesis and should be mentioned and discussed.

4/ In figure, the results are clear but it would be great to have the same axis values for all panels.

5/ The authors use retrograde viral tracing to characterize the inputs to RE. I think that the description of the obtained results do not serve the purpose of the study. There is no statistical quantification and looking at the values described in figure5 panel c I don't understand why the authors choose the LS as target. Furthermore they write the following sentence: "Notably, within the thalamic areas, RE neurons received the most extensive inputs from the lateral septum (LS),

with significantly fewer inputs from the medial septum (MS) (Fig. 5b,c)". This data is not shown. The authors write that optogenetic activation of LS to RE pathways mirror the remimazolam effect on freezing behavior. This sentence is really not clear and the behavioral phenotypes in figure 1 panels i-j and figure 5 panels q-r are really different. The authors should explain the results. Furthermore, by looking at the freezing levels of the vehicle group it seems that the activation of the LS to RE pathways reduces extinction retrieval. The authors should provide the behavioral effects caused by the same manipulation in the AAV-Syn-GFP group. Are we not supposed to see fibers labeled with GFP in the RE in figure 5o?

6/ If I consider freezing levels of all experiments in vehicle conditions. First, the authors show that reducing the expression of the $\gamma 2$ subunit of GABA_A receptors in RE neurons reduces extinction (figure 3n and 3p, vehicle groups, from 20% to 40% freezing on average). Second, the authors claim that activating LS to RE projection enhances extinction (figure 5r, vehicle, 20% freezing on average) but in comparison to what? As mentioned above, there is no group control. Is the AAV-Syn-GFP group at 40%? Third, the authors show that activating vHPC-LS projection receiving RE inputs enhances extinction (figure 6k, from 40% freezing on average for controls to 20% freezing on average for ChR2). Fourth, the authors show that the effect of activating vHPC-LS projection receiving RE inputs on extinction is reversed by chemogenetic inhibition of RE to vHPC projection. To reach this conclusion it took me many reads. I think that the main reason is because of the lack of the behavior of the AAV-Syn-GFP group in figure 5. I think that the effect of remimazolam in this condition is not important and confusing. Are we not supposed to see fibers labeled with mCherry in the LS in figure 6i?

7/ The figure 7 is great and the working model is clear, however the corresponding text lacks the details. mCherry label in figure 7c is green. Also, are we not supposed to see fibers labeled with mCherry in the LS in figure 7c?

Responses to Reviewers' comments (Reviewers' comments in *italic*)

Reviewer #1:

In this paper, the authors investigate the impact of remimazolam, a short-acting benzodiazepine, on fear extinction learning and retrieval, as well as a potential circuit mechanism mediating its effects. First, they observed that systemic remimazolam hindered extinction learning and identified potential brain regions involved through brain-wide cFos screening, including the vHPC, LS, and thalamic RE. Focusing on the RE, the authors find that RE in part mediates remimazolam's effects, with GABA-A receptors in RE being required for its effects. They then relate their cFos quantification findings in the context of an RE-vHPC-LS positive feedback circuit with extensive use of anterograde and retrograde viral vector methods. They find that vHPC-projecting RE neurons require intact GABA-A signaling for proper extinction learning, and restoration recovers the effect of remimazolam on RE. Inhibition of RE from the LS mimics the effects of remimazolam. Connecting these two projections is the vHPC-LS projection, which was found to be largely inhibitory, with activation of it facilitating extinction learning and driving a "positive feedback loop." Overall, the manuscript adds to our understanding of benzodiazepine action and the role of RE circuitry in fear conditioning and extinction. The data is well-presented, and the text is largely written well. The following lists my concerns and suggestions for the authors.

We thank the reviewer for the positive comments.

Major Comments:

1. Knockdown of gabrg2 leads to remimazolam having no effect on RE. The data seem to also suggest that gabrg2 knockdown in RE results in a similar phenotype on extinction learning as remimazolam administration. One would expect that without reliable inhibition from LS due to nonfunctional GABA receptors in this feedback loop, RE neurons would be more activated/excitable. However, Fig. 3f vs 3k in the ACSF condition suggest otherwise. Similarly, Fig. 3m vs 3o suggests that RE is "more" inhibited in vehicle-injected mice. Are RE neurons less excitable following gamma-2

knockdown? How do the authors interpret this? Please clarify this.

We acknowledge the reviewer's concern regarding the seemingly paradoxical hypoexcitability induced by gabrg2 knockdown in RE, despite the expected increase in excitability due to unreliable inhibition from nonfunctional GABA_ARs. We attributed this observation to the potential phenomenon of homeostatic synaptic scaling, which regulates synaptic strength in response to prolonged changes in neuronal activity^{1,2}. In align with the observation of neuronal hypoexcitability caused by gabrg2 knockdown in RE, our behavioral data consistently revealed that mice injected with AAV-CAG-EGFP-U6-gabrg2-ShRNA exhibited significantly higher levels of freezing during fear conditioning, extinction learning, and retrieval compared to those injected with AAV-CAG-EGFP-U6-NC (please see new Supplementary Fig. 4f-i and its legend; page 12, end of the last paragraph and page 13, beginning of the 1st paragraph). This finding supports a negative correlation between RE neuronal excitability and the behavioral manifestation of conditioned fear, consistent with the established physiological role of RE in fear extinction.

To address concerns regarding potential confounding factors arising from chronic genetic knockdown of gabrg2, we included pharmacological data using the benzodiazepine antagonist, flumazenil. Our results showed that flumazenil effectively nullified the enhancing effects of remimazolam on eIPSCs in RE, while also mitigated the undermining effects of RE-specific remimazolam delivery on fear extinction (please see new Supplementary Fig. 3 and its legend; page 11, end of the 2nd paragraph and page 12, beginning of the 1st paragraph). These findings collectively reinforce the notion that remimazolam undermines fear extinction by targeting GABA_ARs.

References

1. Davis GW. Homeostatic control of neural activity: from phenomenology to

molecular design. *Annu Rev Neurosci* 29, 307-323 (2006).

2. Pozo K, Goda Y. Unraveling mechanisms of homeostatic synaptic plasticity. *Neuron* 66, 337-351 (2010).

*2. Are there any effects on learning and memory more generally due to remimazolam?
Add discussion if there are literature.*

Done accordingly (please see page 27, the 2nd paragraph and page 28, the 1st paragraph).

3. That the RE sends excitatory projections to vHPC (line 325-326) is not clear until readers reach the end of the results. The authors may highlight evidence, either from their data or the literature, that show RE-HPC projections are excitatory (or that remimazolam-affected RE neurons are glutamatergic). It would be helpful to include this information earlier on, so that understanding the nature of this circuit is easier.

Rephrased accordingly (please see page 14, middle of the 1st paragraph; page 19, middle of the 2nd paragraph).

4. The methods should be checked to ensure that all information regarding performed experiments is included. They should also be in-depth enough to allow for reproducibility.

Rephrased accordingly (please see the Methods section).

a. Details regarding systemic administration of remimazolam are not present.

Included accordingly (please see the page 32, the last paragraph and page 33, the 1st paragraph).

b. More information related to trials vs blocks in extinction learning should be added. For example, “2-trial block” is not defined or described other than in the figures. Are pairwise comparisons (e.g. as in Fib 1b) between 2-trial-averaged blocks?

In response to the reviewer’s suggestion, we have provided a definition of “2-trial block” in the Methods section (please see the page 31, end of the 2nd paragraph). Based on this definition, pairwise comparisons between 2-trial-averaged blocks were conducted consistently throughout the entire study.

c. What is the rationale for the remimazolam concentrations used (50 μ M in vivo and 100 for electrophysiology)? Any previous literature?

We appreciate the reviewer for highlighting this important concern. Upon careful review of the original data, we have discovered an error in our description of the remimazolam concentrations used across the electrophysiology and behavioral manipulation studies. In fact, the remimazolam concentrations employed in both *in vivo* behavioral and *in vitro* electrophysiology experiments consistently remained at 100 μ M. We apologize for any confusion arising from the earlier version of our manuscript. These inaccuracies have been rectified (please see page 9, middle of the 2nd paragraph) in the revised manuscript.

d. Was the whole RE nucleus used for cFos quantifications? One slice? More details are needed. Does the RE uniformly respond to remimazolam across the anterior-posterior axis?

In accordance with the reviewer’s suggestion, we have elaborated on the c-fos quantification process in the Methods section (please see page 42, the 2nd paragraph and page 43, the 1st paragraph). Additionally, we have included quantification data regarding c-fos-positive neurons along the anterior-posterior axis of the RE (Bregma -0.58 , -0.82 , and -1.06 mm), demonstrating that the

anterior RE exhibits a more significant response to remimazolam compared to the posterior RE (please see new Fig. 1f and page 9, middle of the 1st paragraph).

5. Fig. 5f,g does not support the claim made in the sentence starting on line 259. A direct comparison between the two conditions mentioned might be helpful.

Rephrased accordingly (please see page 17, end of the 1st paragraph and page 18, beginning of the 1st paragraph).

Minor Comments:

1. Some word choices seem out of place.

a. Intermediation as mentioned in the title and abstract could be better worded.

We appreciate the reviewer’s suggestion. After careful consideration, we believe that “intermediation” more accurately conveys the concept we intended to emphasize. In our study, we aimed to elucidate how remimazolam as the ultra-short-acting benzodiazepine affects the transmission of information between the hippocampus and lateral septum within the hippocamposeptal circuit, thereby influencing fear extinction. The term “intermediation” highlights the intermediary role played by remimazolam in the thalamic nucleus reuniens in the communication between the hippocampus and lateral septum, aligning closely with our intended emphasis. Therefore, we are inclined to retain the term “intermediation” in the title and abstract for its relevance to the discussed context. We hope this clarification addresses the reviewer’s concern adequately.

b. “Originates” (line 83) could be better put (sends?)

Rephrased accordingly (please see the page 5, middle of the last paragraph).

c. “Injected” (line 142) would be better put as infused

Rephrased accordingly (please see the page 9, end of the 2nd paragraph).

d. “Chemical genetic” (line 214) should be chemogenetic

Rephrased accordingly (please see the page 15, middle of the 1st paragraph).

2. Some phrases/sentences are unclear or undefined.

a. The sentence beginning on line 65 is difficult to understand.

Rephrased accordingly (please see the page 4, end of the 1st paragraph).

b. Define oIPSCs (line 259)

Rephrased accordingly (please see the page 17, end of the 1st paragraph).

c. The sentence beginning on line 307 should be stated clearer, as it could be a great summary of circuit and its functions as investigated up to that point.

Rephrased accordingly (please see the page 20, the 2nd paragraph).

3. The RE appears to encapsulate the xiphoid nucleus as it’s drawn on Fig. 1e. Ensure that the xiphoid was not used for quantifications and update the ROI on 1e.

Corrected accordingly (please see the new Fig. 1e). Importantly, our quantification of the RE did not include the xiphoid nucleus.

4. It would be helpful to mention in the main text (or show in the figure) that remimazolam was administered systemically/intraperitoneally for Figures 3-5.

Rephrased accordingly (please see the new Fig. 3m, o and 4f, k; page 13, middle of the last paragraph; page 15, middle of the 1st paragraph).

5. EGFP expression for the reversible knockdown as shown in Fig. 4b does not give the impression that the virus expressed. An image comparable to the representative image in Fig. 3b, or a more representative one in general, should be used.

Corrected as suggested. Additionally, we have included a high-magnification image in the revised manuscript demonstrating the co-expression of EGFP and mCherry in the RE (please see the new Fig. 4b and its legend).

6. Fig. 6n's "hM4Di-mCherry" should be red, not green.

Corrected accordingly (please see the new Fig. 6n).

Reviewer #2:

Cheung et al. found that remimazolam potentiated GABA transmission in the RE and suppressed RE neuronal activity, resulting in the alteration of hippocamposeptal circuit function. This mechanism underlies how remimazolam hinders fear extinction. The study is intriguing and holds significant implications for comprehending the functional role and mechanism of remimazolam in fear extinction regulation. However, there is room for improvement in the writing as the logic in the introduction is not very clear. Additionally, some sentences lack clarity in the description. I have a few points listed as below.

We appreciate the positive feedback from the reviewer. Additionally, following the reviewer's suggestions, we conducted additional experiments.

1. Regarding the entire set of figures concerning freezing levels in the Ext and Retr

phases, it is evident that the freezing levels for the vehicle varied widely across different groups (ranging from 20% to 50%). In contrast, the Remimazolam group exhibited a relatively stable freezing level, consistently at around 40-50% during the extinction retrieval session (except for fig1j, nearly 60% for remimazolam group). This discrepancy highlights a notable difference between the vehicle and Remimazolam, primarily attributed to the unstable freezing levels observed in the vehicle group. The authors are encouraged to discuss these results and provide further explanations.

We are grateful to the reviewer for raising this significant concern. Fear extinction, as a behavioral paradigm, is known for its inherent variability, influenced by various uncontrolled factors such as the strength of memory acquisition. Given this variability, maintaining consistent extinction levels across the manuscript poses a challenge, especially at this stage of research. Additionally, the limitations on the number of experimental animals used may contribute to the observed variability in fear extinction control. It is important to emphasize that our control and experimental intervention groups consistently involve the same cohort of animals, ensuring internal validity.

Behavioral data were collected and processed randomly, with all tests and analyses conducted in a blinded manner. Furthermore, all data points were included in the analysis, and replication attempts for behavioral analysis yielded successful outcomes. Despite fluctuations observed in freezing levels within the control groups, both control and experimental groups consistently demonstrated the expected behavioral differences. Therefore, these fluctuations do not undermine the overall conclusions drawn from our study.

In response to the reviewer's suggestion, we have provided further explanations to discuss these results (please see page 29, the 2nd paragraph).

2. Why did the authors choose to use remimazolam at a dosage of 3.75 mg/kg, and how

does this relate to its clinical application?

We apologize for the previous unclear and inaccurate description of the remimazolam dosage in our manuscript. Initially, we stated the dosage as 3.75 mg/kg, based on injecting approximately 0.1 ml of a 1 mg/ml remimazolam solution into mice weighing 26-28 g. We have now corrected this statement to reflect a dosage of approximately 4 mg in 5 ml per kg body weight of mice.

Additionally, we have provided dosage screening results including intraperitoneal (i.p.) administration of doses ranging from 2 to 8 mg/kg. These results showed that doses up to 4 mg/kg of remimazolam did not affect overall locomotor activity in the open field test, while higher doses did. To ensure unaffected detection of conditioned freezing behavior, doses of 2 and 4 mg/kg were selected for subsequent fear extinction studies. Further testing demonstrated that systemic administration of remimazolam at a dose of 4 mg/kg i.p., but not at 2 mg/kg, significantly impeded fear extinction. Therefore, we chose to use a dosage of 4 mg/kg of remimazolam for the subsequent experiments (please see new Supplementary Fig. 1a–i and its legend; page 7, the 1st paragraph; page 8, the 1st paragraph; page 32, the last paragraph and page 33, the 1st paragraph).

3. In Fig1b, the freezing levels observed for remimazolam during the extinction phase are even higher than the last two points seen during the conditioning phase. What could explain this finding? Additionally, if remimazolam administered via i.p. injection affects freezing levels in naïve mice?

We appreciate the reviewer’s thoughtful critique regarding the variability of behavioral data. To address the observed difference in freezing levels between the last two points during the conditioning phase on day 1 and the first points during the extinction phase on day 2, we suggest the overnight consolidation hypothesis as a potential explanation. Additionally, in response to the reviewer’s suggestion,

we have included new findings indicating that remimazolam at 4 mg/kg i.p. did not influence freezing levels in mice subjected to similar conditioning and extinction protocols without foot shocks (CS only) (please see new Supplementary Fig. 1j–l and its legend; page 8, end of the 1st paragraph). These results imply that administering remimazolam is unlikely to impact freezing levels in naïve mice.

4. *What does “two-trial block” mean? Please include the description.*

Included accordingly (please see page 31, end of the 2nd paragraph).

5. *Line 133: dorsal part (LSR) should be LSD.*

Rephrased accordingly (please see page 9, end of the 1st paragraph).

6. *Fig1h, Line142: the images show that only one tract of the cannula above the RE, while the text reads that drug was bilaterally injected into the RE. Any typo?*

We apologize for the inaccurate description. Due to the centrally distributed location of the RE, we conducted a single central injection allowing bilateral infusion. We have now rectified this misleading issue (please see page 9, middle of the 2nd paragraph; page 35, middle of the last paragraph).

7. *Line 143: the use of “on the following day” is inappropriate, it is still referring to the same day.*

Rephrased accordingly (please see the page 10, beginning of the 1st paragraph).

8. *Why did authors use 50 μ M Remimazolam for in vivo behavioral study, while use 100 μ M Remimazolam for slice recordings?*

We appreciate the reviewer for highlighting this important concern. Upon careful review of the original data, we have discovered an error in our description of the remimazolam concentrations used across the electrophysiology and behavioral manipulation studies. In fact, the remimazolam concentrations employed in both *in vivo* behavioral and *in vitro* electrophysiology experiments consistently remained at 100 μ M. We apologize for any confusion arising from the earlier version of our manuscript. These inaccuracies have been rectified (please see page 9, middle of the 2nd paragraph) in the revised manuscript.

9. Fig2b and 2e, Additional sample traces with GABAAR antagonist or glutamate receptor antagonists should be included to confirm that evoked currents are IPSCs or EPSCs.

Done accordingly (please see new Fig. 2b, e and page 10, the 2nd paragraph).

10. Fig2i,3f,3k: Rheobase (mA?), it should be pA.

Corrected accordingly (please see new Figs. 2i, 3f, 3k, 4e, 4h).

11. Fig4b, it is hard to tell the EGFP expression in RE neurons. Please also include a high magnification image to allow the reader to see the expression of EGFP and mCherry in the RE.

Corrected as suggested. We have included a high-magnification image in the revised manuscript demonstrating the co-expression of EGFP and mCherry in the RE (please see the new Fig. 4b and its legend).

12. Line1154, please verify the concentration of CNO used for ephys. 100 μ M maybe too high!

We have verified the concentration of CNO used for electrophysiology and confirmed that it is indeed 10 μ M. We apologize for any confusion arising from the earlier version of our manuscript. These inaccuracies have been rectified (please see the new legends for Fig. 6o and Supplementary Fig. 5h) in the revised manuscript.

13. Line238-240, fig5b,c: “RE neurons received substantial inputs from the midbrain periaqueductal gray (PAG), basolateral amygdala (BLA), and nucleus accumbens (NAc)”. no images or data show BLA and NAc.

We appreciate the reviewer for bringing up this important concern. Upon thorough re-evaluation of the original data, we have identified an error in our previous manuscript. Contrary to what was stated previously, there were negligible inputs to the RE from the basolateral amygdala (BLA) and nucleus accumbens (NAc). We have rectified this error in the revised manuscript (please see page 16, end of the 1st paragraph).

14. Fig5f,i,l: Please include the light application bar aligned with the current trace. What is the latency for oIPSC?

Corrected accordingly (please see new Figs. 5f, i, l). The quantification of the latency for oIPSC was included in new the right panel of new Fig. 5j.

15. Fig5o, please also include a high magnification image to show the expression of GFP fibers are seen in the RE.

Corrected accordingly (please see new Fig. 5o).

16. Please provide clarification in the methods section regarding whether the delivery of the virus or drug to the brain region is bilateral or unilateral, as well as details

regarding the cannula implantation.

Rephrased accordingly (please see page 34, end of the last paragraph and page 35, middle of the 1st paragraph).

17. Fig5q: the labels are marked wrong (vehicle and Remimazolam?)

Corrected accordingly (please see new Fig. 5q).

18. Fig6i, Fig7c, please include a high magnification image to show the expression of mCherry in the LS and RE.

Corrected accordingly (please see new Figs. 6i, 7c).

Reviewer #3:

In the submission "An ultra-short-acting benzodiazepine in thalamic nucleus reuniens undermines fear extinction via intermediation of hippocamptoseptal circuits" the authors address an interesting question, what is the neuronal mechanisms underlying the effect of remimazolam on the learning of conditioned fear extinction? The authors employ appropriate methodologies to answer the question and the experiments are well executed. However, the authors tend to overstate some results and omit some controls and discussion about important aspects of the work. This study has to be revised but definitely deserves publication.

We are grateful for the positive feedback from the reviewer. In response to their valuable suggestions, we have conducted additional experiments.

Please find below my comments for each part/figure.

First of all, concerning the title of the paper. Why do the authors emphasize the fact

that remimazolam is an ultra-short-acting benzodiazepine? They are not showing the dynamic of action of the remimazolam.

We apologize for the inaccurate description regarding the emphasis on the fact that remimazolam is an ultra-short-acting benzodiazepine. Indeed, the anxiolytic effects of benzodiazepines can be intertwined with their sedative, hypnotic, and anesthetic effects, necessitating the use of ultra-short-acting benzodiazepines to delineate their impact on fear extinction while minimizing interference with interoceptive states. Remimazolam, a novel benzodiazepine derivative, has been developed to enhance sedation profiles, featuring a quicker onset, shorter sedation duration, and faster recovery compared to current agents. The well-characterized dynamics of action of remimazolam as an ultra-short-acting benzodiazepine provide an excellent opportunity to address its effects and underlying neural circuit mechanisms on fear extinction (please see page 4, end of the 1st paragraph).

1. The authors aim at characterizing the effect of remimazolam on the learning of conditioned fear extinction. Intraperitoneal and intracerebral injections of remimazolam before extinction have two effects: increasing freezing during extinction in presence of the drug (day2) and during retrieval in absence of the drug (day3). As stated in the title, abstract and text of the paper the authors conclude from these results that remimazolam undermines fear extinction. However, remimazolam does more than that. Vehicle and remimazolam injected mice show different levels of freezing during the first CS presentations of extinction showing an effect of remimazolam on fear expression itself. This result is not interpreted or discussed by the authors.

Data in figure 1 panels c and j are the same. Please provide the right results.

In response to the reviewer's insightful feedback regarding the effects of remimazolam beyond fear extinction, we have included additional interpretation in the revised manuscript (please see page 8, middle of the 1st paragraph). Additionally, we have rectified the error of displaying the same data in Fig. 1c and

1j in the revised manuscript (please see new Fig. 1c, j).

2. Taken together, the results of the c-fos (figure1f) and the patch recordings of RE neurons (figure2), show that remimazolam decreases neuronal excitability in RE neurons probably by the potentiation of GABAergic transmission. The authors use different values for the axis showing rheobase, eIPSC amplitude and neuron excitability throughout the manuscript. This does not help to compare the results between experiments.

Corrected accordingly to ensure consistency in the values displayed on the axes showing rheobase, eIPSC amplitude, and neuron excitability throughout the manuscript (please see new Figs. 2–4).

3. The authors aim at demonstrating that remimazolam potentiates GABAergic transmission in RE. For that they use a shRNA strategy to reduce the expression of the $\gamma 2$ subunit of GABA_A receptors. The authors mention in the text that the used viral strategy is well-verified. They should provide at least histological proof of the efficacy of this virus. In figure 3, the authors compare the effect of remimazolam in presence or absence of $\gamma 2$ subunit of GABA_A receptors within RE. The results are clear, it would be great to have the same axis values for all panels.

Regarding the behavioral effect of such manipulations, the authors provide statistical comparisons of freezing levels only between vehicle and remimazolam for each viral condition. By looking at the data, it seems that AAV-CAG-EGFP-U6-gabrg2-ShRNA-injected and AAV-CAG-EGFP-U6-NC-injected animals exhibit different levels of freezing when comparing vehicle conditions. If it's the case, it means that reducing the expression of the $\gamma 2$ subunit of GABA_A receptors in RE neurons affects fear expression and extinction learning. This result argues in favor of the authors hypothesis and should be mentioned and discussed.

The histological proof of the efficacy of AAV-CAG-EGFP-U6-Gabrg2-ShRNA has

been included in the revised manuscript (please see new Supplementary Fig. 4c–e and page 12, middle of the 2nd paragraph). As suggested, to better compare the effect of remimazolam in the presence or absence of the GABA_AR- γ 2 subunit within RE, we updated the new Fig. 3 with the same axis values for all comparable panels (please see new Fig. 3).

Moreover, regarding the behavioral effect of Gabrg2 knockdown in RE, we included new set data showing that mice injected with AAV-CAG-EGFP-U6-gabrg2-ShRNA exhibited significantly higher levels of freezing during fear conditioning, extinction learning, and retrieval compared to those injected with AAV-CAG-EGFP-U6-NC (please see new Supplementary Fig. 4f–i and its legend; page 12, end of the last paragraph and page 13, the beginning of the 1st paragraph). We attributed this observation to the potential phenomenon of homeostatic synaptic scaling, which regulates synaptic strength in response to prolonged changes in neuronal activity^{1,2}. These findings support a negative correlation between RE neuronal excitability and the behavioral manifestation of conditioned fear, consistent with the established physiological role of RE in fear extinction.

References

1. Davis GW. Homeostatic control of neural activity: from phenomenology to molecular design. *Annu Rev Neurosci* 29, 307-323 (2006).
2. Pozo K, Goda Y. Unraveling mechanisms of homeostatic synaptic plasticity. *Neuron* 66, 337-351 (2010).

4. In figure, the results are clear but it would be great to have the same axis values for all panels.

Corrected to ensure consistency in the values displayed on the axes showing the same items throughout the manuscript (please see new Figs. 2–4).

5. The authors use retrograde viral tracing to characterize the inputs to RE. I think that the description of the obtained results do not serve the purpose of the study. There is no statistical quantification and looking at the values described in figure 5 panel c I don't understand why the authors choose the LS as target. Furthermore they write the following sentence: "Notably, within the thalamic areas, RE neurons received the most extensive inputs from the lateral septum (LS), with significantly fewer inputs from the medial septum (MS) (Fig. 5b,c)". This data is not shown.

The authors write that optogenetic activation of LS to RE pathways mirror the remimazolam effect on freezing behavior. This sentence is really not clear and the behavioral phenotypes in figure 1 panels i-j and figure 5 panels q-r are really different. The authors should explain the results. Furthermore, by looking at the freezing levels of the vehicle group it seems that the activation of the LS to RE pathways reduces extinction retrieval. The authors should provide the behavioral effects caused by the same manipulation in the AAV-Syn-GFP group.

Are we not supposed to see fibers labeled with GFP in the RE in figure 5o?

We appreciate the insightful critique raised by the reviewer. The absence of statistical quantification in Fig. 5c can be attributed to the inherent variability in size and neuron numbers across different brain regions. Additionally, our decision not to perform quantification here was influenced by analogous studies in existing literatures.

The selection of the lateral septum (LS) as the target for studying inputs to the RE is based not only on its abundance as a source of long-range inputs to the RE but also on its specific neuronal properties. LS predominantly contains long-range GABAergic neurons and a smaller population of glutamatergic neurons, likely resulting in distinct change patterns between LS and RE in response to remimazolam treatment. As evidenced by our findings, systemic administration of remimazolam led to enhanced LS activation while simultaneously reducing RE activation during extinction retrieval (please see Fig. 1d-f). This suggests a

potential positive modulation of long-range GABAergic projections from LS to RE by remimazolam in the context of fear extinction (please see page 17, beginning of the 2nd paragraph).

The oversight in stating inputs to RE from both the lateral septum (LS) and medial septum (MS) without corresponding data has been rectified. We have now included the relevant data in the revised manuscript (please see new Fig. 5b, c).

Regarding the optogenetic activation of LS to RE pathways, a typographical error was identified in describing the groups (vehicle versus remimazolam) in Fig. 5q, r, which has been corrected to EGFP versus ChR2. Indeed, compared to the control group injected with AAV-Syn-EGFP, optogenetic activation of LS → RE pathways (AAV-Syn-ChR2-EGFP) significantly hindered fear extinction, mirroring the effects of remimazolam. The apparent difference in extinction kinetics shown in Fig. 1b, c and Fig. 5q, r is likely attributable to batch variation effects and potential additional effects of remimazolam beyond the sole activation of LS → RE pathways.

We apologize for any confusion caused by the oversight in labeling the groups and have provided clarification in the revised manuscript. Additionally, a refined image showing fibers labeled with EGFP in the RE has been included in the new Fig. 5o.

6. If I consider freezing levels of all experiments in vehicle conditions. First, the authors show that reducing the expression of the $\gamma 2$ subunit of GABA_A receptors in RE neurons reduces extinction (figure 3n and 3p, vehicle groups, from 20% to 40% freezing on average). Second, the authors claim that activating LS to RE projection enhances extinction (figure 5r, vehicle, 20% freezing on average) but in comparison to what? As mentioned above, there is no group control. Is the AAV-Syn-GFP group at 40%? Third, the authors show that activating vHPC-LS projection receiving RE inputs enhances

extinction (figure 6k, from 40% freezing on average for controls to 20% freezing on average for Chr2). Fourth, the authors show that the effect of activating vHPC-LS projection receiving RE inputs on extinction is reversed by chemogenetic inhibition of RE to vHPC projection. To reach this conclusion it took me many reads. I think that the main reason is because of the lack of the behavior of the AAV-Syn-GFP group in figure 5. I think that the effect of remimazolam in this condition is not important and confusing. Are we not supposed to see fibers labeled with mCherry in the LS in figure 6i?

Once more, we apologize for the typographical errors concerning the optogenetic activation of LS to RE pathways (Fig. 5q, r), which may have led to a misleading impression of the absence of behavior in the AAV-Syn-GFP group. Indeed, the labeling in the groups (vehicle versus remimazolam) should be replaced with EGFP versus Chr2 (please see our response to the point #6 above).

Additionally, a refined image showing fibers labeled with mCherry in the LS has been included in the new Fig. 6i.

7. The figure 7 is great and the working model is clear, however the corresponding text lacks the details. mCherry label in figure 7c is green. Also, are we not supposed to see fibers labeled with mCherry in the LS in figure 7c?

We have incorporated additional details regarding Figure 7 and the working model into the revised manuscript (please see page 22, end of the 1st paragraph).

The color of the mCherry label in Fig. 7c has been rectified to red. Additionally, we have included an improved image displaying fibers labeled with mCherry in the LS in the new Fig. 7c.

REVIEWERS' COMMENTS:

Reviewer #1 (Remarks to the Author):

The authors have sufficiently addressed my concerns. I think the paper is ready for publication.

Reviewer #2 (Remarks to the Author):

The authors have addressed my concerns, so the manuscript is ready to proceed for publication.

Reviewer #3 (Remarks to the Author):

The authors have conducted a thorough revision of the original manuscript which addresses most of the reviewers' comments.

I am still not convinced by the text associated with the panels a-c of the figure 5 (lines 275-298). Without statistical quantification, the authors cannot conclude the following: "Notably, within the thalamic areas, RE neurons received the most extensive inputs from the lateral septum (LS), with significantly fewer inputs from the medial septum (MS) (Fig. 5b,c)". This is wrong.

I now recommend the paper for publication.

Responses to Reviewers' comments (Reviewers' comments in *italic*)

Reviewer #1:

The authors have sufficiently addressed my concerns. I think the paper is ready for publication.

We thank the reviewer for the positive comments.

Reviewer #2:

The authors have addressed my concerns, so the manuscript is ready to proceed for publication.

We thank the reviewer for the positive comments.

Reviewer #3:

The authors have conducted a thorough revision of the original manuscript which addresses most of the reviewers' comments.

I am still not convinced by the text associated with the panels a-c of the figure 5 (lines 275-298). Without statistical quantification, the authors cannot conclude the following: "Notably, within the thalamic areas, RE neurons received the most extensive inputs from the lateral septum (LS), with significantly fewer inputs from the medial septum (MS) (Fig. 5b,c)". This is wrong.

I now recommend the paper for publication.

We thank the reviewer for the support of publication. Following the reviewer's comments, we have rephrased the sentence in question to accurately reflect the data presented in panels a-c of Figure 5 (please see page 16, middle of the 1st paragraph).